# Stochastic Gradient Descent for Gaussian Processes Done Right

**Jihao Andreas Lin**[*,1,2]  **Shreyas Padhy**[*,1]  **Javier Antorán**[*,1]  **Austin Tripp**[1]
**Alexander Terenin**[1,3]  **Csaba Szepesvári**[4]  **José Miguel Hernández-Lobato**[1]  **David Janz**[4]
[1]University of Cambridge   [2]Max Planck Institute for Intelligent Systems
[3]Cornell University   [4]University of Alberta

## Abstract

As is well known, both sampling from the posterior and computing the mean of the posterior in Gaussian process regression reduces to solving a large linear system of equations. We study the use of stochastic gradient descent for solving this linear system, and show that when *done right*—by which we mean using specific insights from the optimisation and kernel communities—stochastic gradient descent is highly effective. To that end, we introduce a particularly simple *stochastic dual descent* algorithm, explain its design in an intuitive manner and illustrate the design choices through a series of ablation studies. Further experiments demonstrate that our new method is highly competitive. In particular, our evaluations on the UCI regression tasks and on Bayesian optimisation set our approach apart from preconditioned conjugate gradients and variational Gaussian process approximations. Moreover, our method places Gaussian process regression on par with state-of-the-art graph neural networks for molecular binding affinity prediction.

## 1   Introduction

Gaussian process regression is the standard modelling choice in Bayesian optimisation and other applications where uncertainty-aware decision-making is required to gather data efficiently. The main limitation of Gaussian process models is that their fitting requires solving a large linear system of equations which, using direct methods, has a cost cubic in the number of observations.

Standard approaches to reducing the cost of fitting Gaussian process models either apply approximations to reduce the complexity of the linear system, such as the Nyström and related variational approximations (Williams and Seeger, 2000; Titsias, 2009; Hensman et al., 2013), or use carefully engineered iterative solvers, such as preconditioned conjugate gradients (Wang et al., 2019), or employ a combination of both (Rudi et al., 2017). An alternative approach that we focus on in this work is the use of stochastic gradient descent to minimise an associated quadratic objective.

Multiple works have previously investigated the use of SGD with Gaussian process and related kernel models (Lin et al., 2023; Dai et al., 2014; Kivinen et al., 2004), with Lin et al. (2023) pointing out, in particular, that SGD may be competitive with conjugate gradients (CG) when the compute budget is limited, in terms of both the mean and uncertainty predictions that it produces. In this work, we go a step further, and demonstrate that when done right, SGD can outperform CG.

To that end, we propose a simple SGD-based algorithm, which we call *stochastic dual descent* (SDD). Our algorithm is an adaptation of ideas from the stochastic dual coordinate ascent (SDCA) algorithm of Shalev-Shwartz and Zhang (2013) to the large-scale deep-learning-type gradient descent setting, combined with insights on stochastic approximation from Dieuleveut et al. (2017) and Varre et al. (2021). We provide the following evidence supporting the strength of SDD:

1. On standard UCI regression benchmarks with up to 2 million observations, stochastic dual descent either matches or improves upon the performance of conjugate gradients.

---

*Equal contribution.
Code available at: HTTPS://GITHUB.COM/CAMBRIDGE-MLG/SGD-GP.

2. On the large-scale Bayesian optimisation task considered by Lin et al. (2023), stochastic dual descent is shown to be superior to their stochastic gradient descent method and other baselines, both against the number of iterations and against wall-clock time.

3. On a molecular binding affinity prediction task, the performance of Gaussian process regression with stochastic dual descent matches that of state-of-the-art graph neural networks.

In short, we show that a simple-but-well-designed stochastic gradient method for Gaussian processes can be very competitive with other approaches for Gaussian processes, and may make Gaussian processes competitive with graph neural networks on tasks on which the latter are state-of-the-art.

## 2 GAUSSIAN PROCESS REGRESSION

We consider Gaussian process regression over a domain $\mathcal{X} \subset \mathbb{R}^d$, assuming a Gaussian process prior induced by a continuous, bounded, positive-definite kernel $k : \mathcal{X} \times \mathcal{X} \to \mathbb{R}$. Our goal is to, given some observations, either (i) sample from the Gaussian process posterior, or (ii) compute its mean.

To formalise these goals, let some observed inputs $x_1, \ldots, x_n \in \mathcal{X}$ be collected in a matrix $X \in \mathbb{R}^{n \times d}$ and let $y \in \mathbb{R}^n$ denote the corresponding observed real-valued targets. We say that a random function $f_n : \mathcal{X} \to \mathbb{R}$ is a sample from the *posterior Gaussian process* associated with the kernel $k$, data set $(X, y)$, and likelihood variance parameter $\lambda > 0$, if the distribution of any finite set of its marginals is jointly multivariate Gaussian, and the mean and covariance between the marginals of $f_n$ at any $a$ and $a'$ in $\mathcal{X}$ are given by

$$m_n(a) = \mathbb{E}[f_n(a)] = k(a, X)(K + \lambda I)^{-1} y \quad \text{and}$$
$$\mathrm{Cov}(f_n(a), f_n(a')) = k(a, a') - k(a, X)(K + \lambda I)^{-1} k(X, a')$$

respectively, where $K \in \mathbb{R}^{n \times n}$ is the $n \times n$ matrix whose $(i, j)$ entry is given by $k(x_i, x_j)$, $I$ is the $n \times n$ identity matrix, $k(a, X)$ is a row vector in $\mathbb{R}^n$ with entries $k(a, x_1), \ldots, k(a, x_n)$ and $k(X, a')$ is a column vector defined likewise (per the notation of Rasmussen and Williams, 2006).

Throughout, for two vectors $a, b \in \mathbb{R}^p$, with $p \in \mathbb{N}^+$, we write $a^\mathsf{T} b$ for their usual inner product and $\|a\|$ for the 2-norm of $a$. For a symmetric positive semidefinite matrix $M \in \mathbb{R}^{p \times p}$, we write $\|a\|_M$ for the $M$-weighted 2-norm of $a$, given by $\|a\|_M^2 = a^\mathsf{T} M a$. For a symmetric matrix $S$, we write $\|S\|$ for the operator norm of $S$, and $\lambda_i(S)$ for the $i$th largest eigenvalue of $S$, such that $\lambda_1(S) = \|S\|$.

## 3 STOCHASTIC DUAL DESCENT FOR REGRESSION AND SAMPLING

We show our proposed algorithm, *stochastic dual descent*, in Algorithm 1. The algorithm can be used for two purposes: regression, where the goal is to produce a good approximation to the posterior mean function $m_n$, and sampling, that is, drawing a sample from the Gaussian process posterior.

The algorithm takes as input a kernel matrix $K$, the entries of which can be computed 'on demand' using the kernel function $k$ and the observed inputs $x_1, \ldots, x_n$ as needed, a likelihood variance $\lambda > 0$, and a vector $b \in \mathbb{R}^n$. It produces a vector of coefficients $\overline{\alpha}_T \in \mathbb{R}^n$, which approximates

$$\alpha^\star(b) = (K + \lambda I)^{-1} b. \tag{1}$$

To translate this into our goals of mean estimation and sampling, given a vector $\alpha \in \mathbb{R}^n$, let

$$h_\alpha(\cdot) = \sum_{i=1}^n \alpha_i k(x_i, \cdot).$$

Then, $h_{\alpha^\star(y)}$ gives the mean function $m_n$, and thus running the algorithm with the targets $y$ as the vector $b$ can be used to estimate the mean function. Moreover, given a sample $f_0$ from the Gaussian process prior associated with $k$, and noise $\zeta \sim \mathcal{N}(0, \lambda I)$, we have that

$$f_0 + h_{\alpha^\star(y - (f_0(X) + \zeta))}$$

is a sample from the Gaussian process posterior (Wilson et al., 2020, 2021). In practice, one might approximate $f_0$ using random features (as done in Wilson et al., 2020, 2021; Lin et al., 2023).

---

**Algorithm 1** *Stochastic dual descent* for approximating $\alpha^\star(b) = (K + \lambda I)^{-1}b$

---

**Require:** Kernel matrix $K$ with rows $K_1, \ldots, K_n \in \mathbb{R}^n$, targets $b \in \mathbb{R}^n$, likelihood variance $\lambda > 0$,
            number of steps $T \in \mathbb{N}^+$, batch size $B \in \{1, \ldots, n\}$, step size $\beta > 0$,
            momentum parameter $\rho \in [0, 1)$, averaging parameter $r \in (0, 1]$

1:   $v_0 = 0$; $\alpha_0 = 0$; $\overline{\alpha}_0 = 0$                                                ▷ *all in $\mathbb{R}^n$*
2:   **for** $t \in \{1, \ldots, T\}$ **do**
3:       Sample $\mathcal{I}_t = (i_1^t, \ldots, i_B^t) \sim \text{Uniform}(\{1, \ldots, n\})$ independently     ▷ *random coordinates*
4:       $g_t = \frac{n}{B} \sum_{i \in \mathcal{I}_t} ((K_i + \lambda e_i)^\mathsf{T}(\alpha_{t-1} + \rho v_{t-1}) - b_i)e_i$            ▷ *gradient estimate*
5:       $v_t = \rho v_{t-1} - \beta g_t$                                               ▷ *velocity update*
6:       $\alpha_t = \alpha_{t-1} + v_t$                                             ▷ *parameter update*
7:       $\overline{\alpha}_t = r\alpha_t + (1 - r)\overline{\alpha}_{t-1}$                                 ▷ *iterate averaging*
8:   **return** $\overline{\alpha}_T$

---

The SDD algorithm is distinguished from variants of SGD used in the context of GP regression by the following features: (i) SGD uses a dual objective in place of the usual kernel ridge regression objective, (ii) it uses stochastic approximation entirely via random subsampling of the data instead of random features, (iii) it uses Nesterov's momentum, and (iv) it uses geometric, rather than arithmetic, iterate averaging. In the following subsections, we examine and justify each of the choices behind the algorithm design, and illustrate these on the UCI data set POL (Dua and Graff, 2017), chosen for its small size, which helps us to compare against less effective alternatives.

### 3.1   GRADIENT DESCENT: PRIMAL VERSUS DUAL OBJECTIVES

For any $b \in \mathbb{R}^n$, computing the vector $\alpha^\star(b)$ of equation (1) using Cholesky factorisation takes on the order of $n^3$ operations. We now look at how $\alpha^\star(b)$ may be approximated using gradient descent.

As is well known, the vector $\alpha^\star(b)$ is the minimiser of the kernel ridge regression objective,

$$L(\alpha) = \frac{1}{2}\|b - K\alpha\|^2 + \frac{\lambda}{2}\|\alpha\|_K^2$$

over $\alpha \in \mathbb{R}^n$ (Smola and Schölkopf, 1998). We will refer to $L$ as the *primal* objective. Consider using gradient descent to minimise $L(\alpha)$. This entails constructing a sequence $(\alpha_t)_t$ of elements in $\mathbb{R}^n$, which we initialise at the standard but otherwise arbitrary choice $\alpha_0 = 0$, and setting

$$\alpha_{t+1} = \alpha_t - \beta\nabla L(\alpha_t),$$

where $\beta > 0$ is a step-size and $\nabla L$ is the gradient function of $L$. Recall that the speed at which $\alpha_t$ approaches $\alpha^\star(b)$ is determined by the condition number of the Hessian: the larger the condition number, the slower the convergence speed (Boyd and Vandenberghe, 2004). The intuitive reason for this correspondence is that, to guarantee convergence, the step-size needs to scale inversely with the largest eigenvalue of the Hessian, while progress in the direction of an eigenvector underlying an eigenvalue is governed by the step-size multiplied with the corresponding eigenvalue. With that in mind, the *primal* gradient and Hessian are

$$\nabla L(\alpha) = K(\lambda\alpha - b + K\alpha) \quad \text{and} \quad \nabla^2 L(\alpha) = K(K + \lambda I) \tag{2}$$

respectively, and therefore the relevant eigenvalues are bounded as

$$0 \leq \lambda_n(K(K + \lambda I)) \leq \lambda_1(K(K + \lambda I)) \leq \kappa n(\kappa n + \lambda),$$

where $\kappa = \sup_{x \in \mathcal{X}} k(x, x)$ is finite by assumption. These bounds only allow for a step-size $\beta$ on the order of $(\kappa n(\kappa n + \lambda))^{-1}$. Moreover, since the minimum eigenvalue is not bounded away from zero, we do not have a priori guarantees for the performance of gradient descent using $L$.

Consider, instead, minimising the *dual objective*

$$L^*(\alpha) = \frac{1}{2}\|\alpha\|_{K+\lambda I}^2 - \alpha^\mathsf{T}b. \tag{3}$$

The dual $L^*$ has the same unique minimiser as $L$, namely $\alpha^\star(b)$, and the two are, up to rescaling, a strong dual pair (in a sense made precise in Appendix A). The *dual gradient* and Hessian are given by

$$g(\alpha) := \nabla L^*(\alpha) = \lambda\alpha - b + K\alpha \quad \text{and} \quad \nabla^2 L^*(\alpha) = K + \lambda I. \tag{4}$$

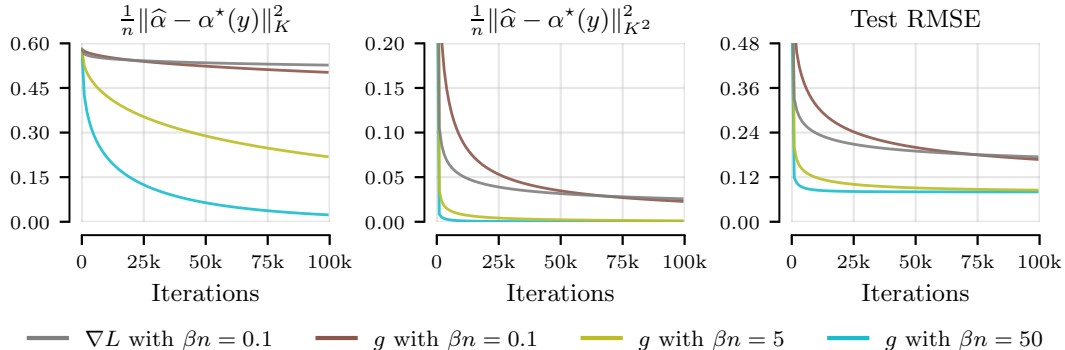

Figure 1: Comparison of full-batch primal and dual gradient descent on POL with varying step-sizes. Primal gradient descent becomes unstable and diverges for $\beta n$ greater than 0.1. Dual gradient descent is stable with larger step-sizes, allowing for markedly faster convergence than the primal. For $\beta n = 0.1$, the dual method makes more progress in the $K$-norm, whereas the primal in the $K^2$-norm.

Examining the eigenvalues of the above Hessian, we see that gradient descent on $L^*$ (dual gradient descent) may use a step-size of up to an order of $\kappa n$ higher than that on $L$ (primal gradient descent), leading to faster convergence. Moreover, since the condition number of the dual satisfies $\mathrm{cond}(K + \lambda I) \leq 1 + \kappa n/\lambda$ and $\lambda$ is positive, we can provide an a priori bound on the number of iterations required for dual gradient descent to convergence to any fixed error level for a length $n$ sequence of observations.

In Figure 1, we plot the results of an experimental comparison of primal and dual gradient descent on the UCI POL regression task. There, gradient descent with the primal objective is stable up to $\beta n = 0.1$ but diverges for larger step-sizes. In contrast, gradient descent with the dual objective is stable with a step-size as much as $500\times$ higher, converges faster and reaches a better solution; see also Appendix B for more detailed plots and recommendations on setting step-sizes. We show this on three evaluation metrics: distance to $\alpha^\star(y)$ measured in $\|\cdot\|_K^2$, the $K$-norm (squared) and in $\|\cdot\|_{K^2}^2$, the $K^2$-norm (squared), and test set root-mean-square error (RMSE). To understand the difference between the two norms, note that the $K$-norm error bounds the error of approximating $h_{\alpha^\star(b)}$ uniformly. Indeed, as shown in Appendix A, we have the bound

$$\|h_\alpha - h_{\alpha^\star(b)}\|_\infty \leq \sqrt{\kappa}\|\alpha - \alpha^\star(b)\|_K, \tag{5}$$

where recall that $\kappa = \sup_{x \in \mathcal{X}} k(x, x)$. Uniform norm guarantees of this type are crucial for sequential decision-making tasks, such as Bayesian optimisation, where test input locations may be arbitrary. The $K^2$-norm metric, on the other hand, reflects the training error.

Examining the two gradients, its immediate that the primal gradient optimises for the $K^2$-norm, while the dual for the $K$-norm. And indeed, we see in Figure 1 that when both methods use $\beta n = 0.1$, up to 70k iterations, the dual method is better on the $K$-norm metric and the primal on $K^2$. Later, the dual gradient method performs better on all metrics. This, too, is to be expected, as the minimum eigenvalue of the Hessian of the dual loss is higher than that of the primal loss.

### 3.2 RANDOMISED GRADIENTS: RANDOM FEATURES VERSUS RANDOM COORDINATES

To compute either the primal or the dual gradients, presented in equations (2) and (4), we need to compute matrix-vector products of the form $K\alpha$, which requires order $n^2$ computations. We now introduce and contrast two types of stochastic approximation for our dual gradient $g(\alpha)$ that reduce the cost to order $n$ per-step, and carefully examine the noise they introduce into the gradient.

For the sake of exposition (and exposition only, this is not an assumption of our algorithm), assume that we are in an $m$-dimensional (finite) linear model setting, such that $K$ is of the form $\sum_{j=1}^m z_j z_j^\mathsf{T}$, where $z_j \in \mathbb{R}^n$ collects the values of the $j$th feature of the observations $x_1, \ldots, x_n$. Then, for $j \sim \mathrm{Uniform}(\{1, \ldots, m\})$, we have that $\mathbb{E}[m z_j z_j^\mathsf{T}] = K$, and therefore

$$\widetilde{g}(\alpha) = \lambda\alpha - b + m z_j z_j^\mathsf{T}\alpha$$

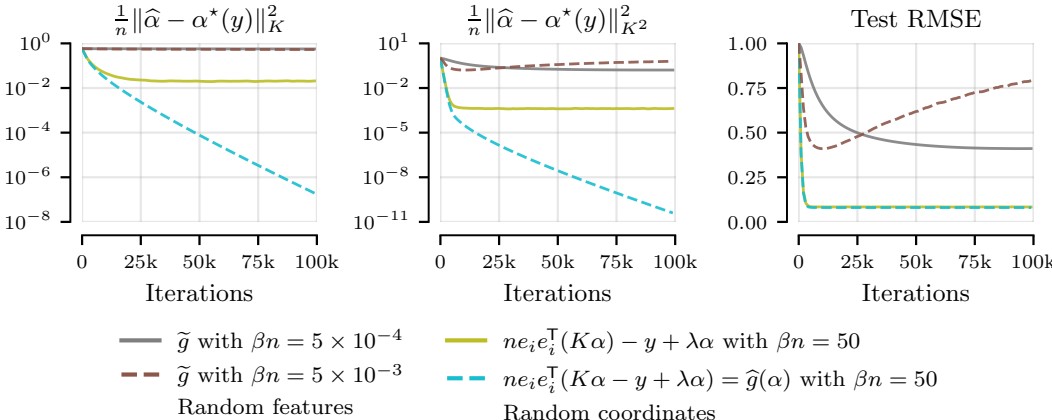

Figure 2: A comparison of dual stochastic gradient descent on the POL data set with either random Fourier features or random coordinates, using batch size $B = 512$, momentum $\rho = 0.9$ and averaging parameter $r = 0.001$ (see Section 3.3 for explanation of latter two). Random features converge with $\beta n = 5 \times 10^{-4}$ but perform poorly, and diverge with a higher step-size. Random coordinates are stable with $\beta n = 50$ and show much stronger performance on all metrics. We include a version of random coordinates where only the $K\alpha$ term is subsampled: this breaks the multiplicative noise property, and results in an estimate which is worse on both the $K$-norm and the $K^2$-norm metric.

is an unbiased estimate of $g(\alpha)$; we call $\widetilde{g}$ the random feature estimate of $g$. The alternative we consider is random coordinates, where we take $i \sim \mathrm{Uniform}(\{1, \ldots, n\})$, and use

$$\widehat{g}(\alpha) = n e_i e_i^\mathsf{T} g(\alpha) = n e_i (\lambda \alpha_i - b_i + K_i \alpha).$$

Observe that $\widehat{g}(\alpha)$ zeros all but the $i$th (of $n$) coordinate of $g(\alpha)$, and then scales the result by $n$; since $\mathbb{E}[n e_i e_i^\mathsf{T}] = I$, $\widehat{g}(\alpha)$ is also an unbiased estimate of $g(\alpha)$. Note that the cost of calculating either $\widetilde{g}(\alpha)$ or $\widehat{g}(\alpha)$ is linear in $n$, and therefore both achieve our goal of reduced computation time.

However, while $\widetilde{g}$ and $\widehat{g}$ may appear similarly appealing, the nature of the noise introduced by these, and thus their qualities, are quite different. In particular, one can show that

$$\|\widehat{g}(\alpha) - g(\alpha)\| \leq \|(M - I)(K + \lambda I)\| \|\alpha - \alpha^\star(b)\|,$$

where $M = n e_i e_i$ is the random coordinate approximation to the identity matrix. As such, the noise introduced by $\widehat{g}(\alpha)$ is proportional to the distance between the current iterate $\alpha$, and the target $\alpha^\star$, making it a so-called *multiplicative noise* gradient estimate (Dieuleveut et al., 2017). Intuitively, multiplicative noise estimates automatically reduce the amount of noise injected as the iterates get closer to their target. On the other hand, for $\widetilde{g}(\alpha)$, letting $\widetilde{K} = m z_j z_j^\mathsf{T}$, we have that

$$\|\widetilde{g}(\alpha) - g(\alpha)\| = \|(\widetilde{K} - K)\alpha\|,$$

and thus the error in $\widetilde{g}(\alpha)$ is *not reduced* as $\alpha$ approaches $\alpha^\star(b)$. The behaviour of $\widetilde{g}$ is that of an *additive noise* gradient estimate. Algorithms using multiplicative noise estimates, when convergent, often yield better performance (see Varre et al., 2021).

Another consideration is that when $m$, the number of features, is larger than $n$, individual features $z_j$ may also be less informative than individual gradient coordinates $e_i e_i^\mathsf{T} g(\alpha)$. Thus, picking features uniformly at random, as in $\widetilde{g}$, may yield a poorer estimate. While this can be addressed by introducing an importance sampling distribution for the features $z_j$ (as per Li et al., 2019), doing so adds implementation complexity, and could be likewise done to improve the random coordinate estimate.

In Figure 2, we compare variants of stochastic dual descent with either random (Fourier) features or random coordinates. We see that random features, which produce high-variance additive noise, can only be used with very small step-sizes and have poor asymptotic performance. We test two versions of random coordinates: $\widehat{g}(\alpha)$, where, as presented, we subsample the whole gradient, and an alternative, $n e_i e_i^\mathsf{T}(K\alpha) - y - \lambda\alpha$, where only the $K\alpha$ term is subsampled. While both are stable with much higher step-sizes than random features, the latter has worse asymptotic performance. This is a

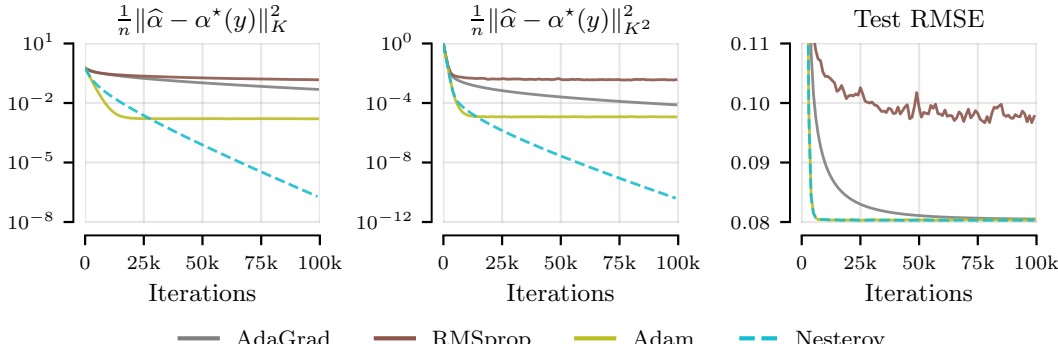

Figure 3: Comparison of dual stochastic gradient descent on the POL data set with different accelera­tion methods, using batch size $B = 512$, a geometric averaging parameter $r = 0.001$, and step-sizes tuned individually for each method (AdaGrad $\beta n = 10$; RMSprop & Adam $\beta n = 0.05$; Nesterov's momentum $\beta n = 50$). Both Adam and Nesterov's momentum perform well on Test RMSE, but the latter performs better on the $K$ and $K^2$ norms.

kind of *Rao-Blackwellisation trap:* introducing the known value of $-y + \lambda\alpha$ in place of its estimate $n e_i e_i^\mathsf{T}(-y + \lambda\alpha)$ destroys the multiplicative property of the noise, making things worse, not better.

Algorithm 1 combines a number of samples into a minibatched estimate to further reduce variance. We discuss the behaviour of different stochastic gradient estimates under minibatching in Appendix B.

There are many other possibilities for constructing randomised gradient estimates. For example, Dai et al. (2014) applied the random coordinate method on top of the random feature method, in an effort to further reduce the order $n$ cost per iteration of gradient computation. However, based on the above discussion, we generally recommend estimates that produce multiplicative noise, and our randomised coordinate gradient estimates as the simplest of such estimates.

### 3.3 NESTEROV'S MOMENTUM AND POLYAK-RUPPERT ITERATE AVERAGING

Momentum, or acceleration, is a range of modifications to the usual gradient descent updates that aim to improve the rate of convergence, in particular with respect to its dependence on the curvature of the optimisation problem (Polyak, 1964). Many schemes for momentum exist, with AdaGrad (Duchi et al., 2011), RMSProp (Tieleman and Hinton, 2012) and Adam (Kingma and Ba, 2015) particularly popular in the deep learning literature. These, however, are designed to adapt to changing curvatures. As our objective has a constant curvature, we recommend the use of the simpler Nesterov's momentum (Nesterov, 1983; Sutskever et al., 2013), as supported by our empirical results in Figure 3, with further results in Figure 4, showing that momentum is vital for the POL data set. The precise form of the updates used is shown in Algorithm 1; we use a momentum of $\rho = 0.9$ throughout.

Polyak-Ruppert averaging returns the average of the (tail of the) stochastic gradient iterates, rather than the final iterate, so as to reduce noise in the estimate (Polyak, 1990; Ruppert, 1988; Polyak and Juditsky, 1992). While Polyak-Ruppert averaging is necessary with constant step-size and additive noise, it is not under multiplicative noise (Varre et al., 2021), and indeed can slow convergence. For our problem, we recommend using *geometric averaging* instead, where we let $\overline{\alpha}_0 = \alpha_0$ and, at each step compute

$$\overline{\alpha}_t = r\alpha_t + (1 - r)\overline{\alpha}_{t-1} \quad \text{for an averaging parameter} \quad r \in (0, 1),$$

and return as our estimate $\overline{\alpha}_T$. Geometric averaging is an anytime approach, in that it does not rely on fixed tail-window size; it can thus be used in combination with early stopping, and the value of $r$ can be tuned adaptively. Figure 4 shows that geometric averaging outperforms both arithmetic averaging and returning the last iterate, $\alpha_T$. Here and throughout, we set $r = 100/T$.

### 3.4 CONNECTIONS TO THE LITERATURE

The dual formulation for the kernel ridge regression objective was first pointed out in the kernel literature in Saunders et al. (1998). It features in textbooks on the topic (see equations (10.95) and

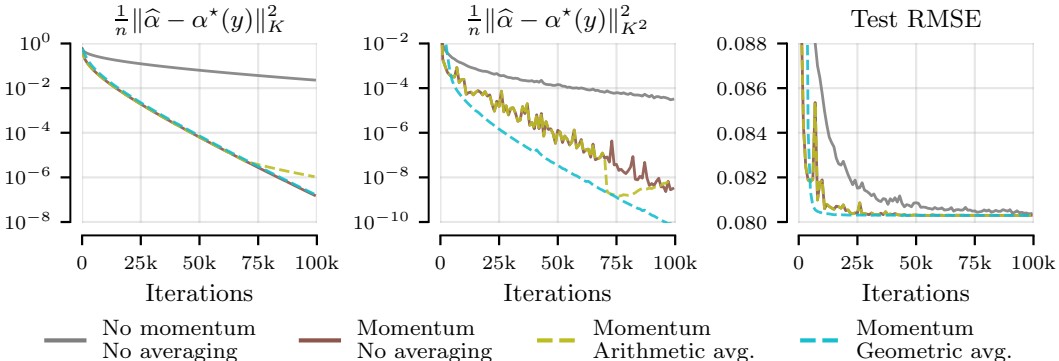

Figure 4: Comparison of optimisation strategies for random coordinate estimator of the dual objective on the POL data set, using momentum $\rho = 0.9$, averaging parameter $r = 0.001$, batch size $B = 128$, and step-size $\beta n = 50$. Nesterov's momentum significantly improves convergence speed across all metrics. The dashed olive line, marked *arithmetic averaging*, shows the regular iterate up until 70k steps, at which point averaging commences and the averaged iterate is shown. Arithmetic iterate averaging slows down convergence in $K$-norm once enabled. Geometric iterate averaging, on the other hand, outperforms arithmetic averaging and unaveraged iterates throughout optimisation.

(10.98) in Smola and Schölkopf, 1998), albeit with no mention of its better conditioning. Gradient descent on the dual objective is equivalent to applying the *kernel adatron* update rule of Frie et al. (1998), stated there for the hinge loss, and to the *kernel least-mean-square* algorithm of Liu et al. (2008), analysed theoretically in Dieuleveut et al. (2017). Shalev-Shwartz and Zhang (2013) introduce *stochastic dual coordinate ascent* (SDCA), which uses the dual objective with random coordinates and analytic line search, and provide convergence results. Tu et al. (2016) implement a block (minibatched) version of SDCA. Bo and Sminchisescu (2008) propose a method similar to SDCA, but with coordinates chosen by active learning. Wu et al. (2023) reimplement the latter two for Gaussian process regression, and link the algorithms to the method of alternating projections.

From the perspective of stochastic gradient descent, the work closest to ours is that of Lin et al. (2023); which we refer to as SGD, to contrast with our SDD. SGD uses the gradient estimate

$$\nabla L(\alpha) \approx nK_i(K_i^\mathsf{T}\alpha - b_i) + \lambda \sum_{j=1}^m z_j z_j^\mathsf{T}\alpha \,, \tag{6}$$

where $i \sim \mathrm{Uniform}(\{1, \ldots, n\})$ is a random index and $\sum_{j=1}^m z_j z_j^\mathsf{T}$ is a random Fourier feature approximation of $K$. This can be seen as targeting the primal loss, with a mixed multiplicative-additive objective. Like SDD, SGD uses geometric iterate averaging and Nesterov's momentum; unlike SDD, the SGD algorithm requires gradient clipping to control the gradient noise. Note that Lin et al. (2023) also consider a modified notion of convergence which involves subspaces of the linear system coefficients. This is of interest in situations where the number of iterations is sufficiently small relative to data size, and generalization properties are important. While our current work does not explicitly cover this setting, it could be studied in a similar manner.

Stochastic gradient descent approaches were also used in similar contexts by, amongst others, Dai et al. (2014) for kernel regression with the primal objective and random Fourier features; Kivinen et al. (2004) for online learning with the primal objective with random sampling of training data; and Antorán et al. (2023) for large-scale Bayesian linear models with the dual objective.

## 4  EXPERIMENTS AND BENCHMARKS

We present three experiments that confirm the strength of our SDD algorithm on standard benchmarks. The first two experiments, on UCI regression and large-scale Bayesian optimisation, replicate those of Lin et al. (2023), and compare against SGD (Lin et al., 2023), CG (Wang et al., 2019) and SVGP (Hensman et al., 2013). Unless indicated otherwise, we use the code, setup and hyperparameters of Lin et al. (2023). Our third experiment tests stochastic dual descent on five molecule-protein binding

Table 1: Root-mean-square error (RMSE), compute time (on an A100 GPU), and negative log-likelihood (NLL), for 9 UCI regression tasks for all methods considered. We report mean values and standard error across five 90%-train 10%-test splits for all data sets, except the largest, where three splits are used. Targets are normalised to zero mean and unit variance. This work denoted by SDD*.

| | | POL 15k | ELEVATORS 17k | BIKE 17k | PROTEIN 46k | KEGGDIR 49k | 3DROAD 435k | SONG 515k | BUZZ 583k | HOUSEELEC 2M |
|---|---|---|---|---|---|---|---|---|---|---|
| | Data Size | | | | | | | | | |
| RMSE | SDD* | **0.08 ± 0.00** | **0.35 ± 0.00** | **0.04 ± 0.00** | **0.50 ± 0.01** | **0.08 ± 0.00** | **0.04 ± 0.00** | **0.75 ± 0.00** | **0.28 ± 0.00** | **0.04 ± 0.00** |
| | SGD | 0.13 ± 0.00 | 0.38 ± 0.00 | 0.11 ± 0.00 | 0.51 ± 0.00 | 0.12 ± 0.00 | 0.11 ± 0.00 | 0.80 ± 0.00 | 0.42 ± 0.01 | 0.09 ± 0.00 |
| | CG | **0.08 ± 0.00** | **0.35 ± 0.00** | **0.04 ± 0.00** | **0.50 ± 0.00** | **0.08 ± 0.00** | 0.18 ± 0.02 | 0.87 ± 0.05 | 1.88 ± 0.19 | 0.87 ± 0.14 |
| | SVGP | 0.10 ± 0.00 | 0.37 ± 0.00 | 0.08 ± 0.00 | 0.57 ± 0.00 | 0.10 ± 0.00 | 0.47 ± 0.01 | 0.80 ± 0.00 | 0.32 ± 0.00 | 0.12 ± 0.00 |
| Time (min) | SDD* | 1.88 ± 0.01 | 1.13 ± 0.02 | 1.15 ± 0.02 | 1.36 ± 0.01 | 1.70 ± 0.00 | **3.32 ± 0.01** | 185 ± 0.56 | 207 ± 0.10 | 47.8 ± 0.02 |
| | SGD | 2.80 ± 0.01 | 2.07 ± 0.03 | 2.12 ± 0.04 | 2.87 ± 0.01 | 3.30 ± 0.12 | 6.68 ± 0.02 | 190 ± 0.61 | 212 ± 0.15 | 69.5 ± 0.06 |
| | CG | **0.17 ± 0.00** | **0.04 ± 0.00** | **0.11 ± 0.01** | **0.16 ± 0.01** | **0.17 ± 0.00** | 13.4 ± 0.01 | 192 ± 0.77 | 244 ± 0.04 | 157 ± 0.01 |
| | SVGP | 11.5 ± 0.01 | 11.3 ± 0.06 | 11.1 ± 0.02 | 11.1 ± 0.02 | 11.5 ± 0.04 | 152 ± 0.15 | 213 ± 0.13 | 209 ± 0.37 | 154 ± 0.12 |
| NLL | SDD* | **-1.18 ± 0.01** | **0.38 ± 0.01** | -2.49 ± 0.09 | **0.63 ± 0.02** | **-0.92 ± 0.11** | **-1.70 ± 0.01** | **1.13 ± 0.01** | **0.17 ± 0.06** | **-1.46 ± 0.10** |
| | SGD | -0.70 ± 0.02 | 0.47 ± 0.00 | -0.48 ± 0.08 | 0.64 ± 0.01 | -0.62 ± 0.07 | -0.60 ± 0.00 | 1.21 ± 0.00 | 0.83 ± 0.07 | -1.09 ± 0.04 |
| | CG | **-1.17 ± 0.01** | **0.38 ± 0.00** | **-2.62 ± 0.06** | 0.62 ± 0.01 | **-0.92 ± 0.10** | 16.3 ± 0.45 | 1.36 ± 0.07 | 2.38 ± 0.08 | 2.07 ± 0.58 |
| | SVGP | -0.67 ± 0.01 | 0.43 ± 0.00 | -1.21 ± 0.01 | 0.85 ± 0.01 | -0.54 ± 0.02 | 0.60 ± 0.00 | 1.21 ± 0.00 | 0.22 ± 0.03 | -0.61 ± 0.01 |

Table 2: Test set $R^2$ scores obtained for each target protein on the DOCKSTRING molecular binding affinity prediction task. Results with $(\cdot)^\dagger$ are from García-Ortegón et al. (2022), those with $(\cdot)^\ddagger$ are from Tripp et al. (2023). SVGP uses 1000 inducing points. SDD* denotes this work.

| Method | ESR2 | F2 | KIT | PARP1 | PGR |
|---|---|---|---|---|---|
| Attentive FP[†] | **0.627** | **0.880** | **0.806** | **0.910** | **0.678** |
| MPNN[†] | 0.506 | 0.798 | 0.755 | 0.815 | 0.324 |
| XGBoost[†] | 0.497 | 0.688 | 0.674 | 0.723 | 0.345 |

| Method | ESR2 | F2 | KIT | PARP1 | PGR |
|---|---|---|---|---|---|
| SDD* | **0.627** | **0.880** | 0.790 | 0.907 | 0.626 |
| SGD | 0.526 | 0.832 | 0.697 | 0.857 | 0.408 |
| SVGP[‡] | 0.533 | 0.839 | 0.696 | 0.872 | 0.477 |

affinity prediction benchmarks of García-Ortegón et al. (2022). We include detailed descriptions of all three experimental set-ups and some additional results in Appendix C.

## 4.1 UCI REGRESSION BASELINES

We benchmark on the 9 UCI repository regression data sets (Dua and Graff, 2017) used by Wang et al. (2019) and Lin et al. (2023). We run SDD for 100k iterations, the same number used by Lin et al. (2023) for SGD, but with step-sizes $100\times$ larger than Lin et al. (2023), except for ELEVATORS, KEGGDIRECTED, and BUZZ, where this causes divergence, and we use $10\times$ larger step-sizes instead. We run CG to a tolerance of $0.01$, except for the 4 largest data sets, where we stop CG after 100 iterations—this still provides CG with a larger compute budget than first-order methods. CG uses a pivoted Cholesky preconditioner of rank $100$. For SVGP, we use $3,000$ inducing points for the smaller five data sets and $9,000$ for the larger four, so as to match the runtime of the other methods.

The results, reported in Table 1, show that SDD matches or outperforms all baselines on all UCI data sets in terms of root-mean-square error of the mean prediction across test data. SDD strictly outperforms SGD on all data sets and metrics, matches CG on the five smaller data sets, where the latter reaches tolerance, and outperforms CG on the four larger data sets. The same holds for the negative log-likelihood metric (NLL), computed using $64$ posterior function samples, except on BIKE, where CG marginally outperforms SDD. Since SDD requires only one matrix-vector multiplication per step, as opposed to two for SGD, it provides about $30\%$ wall-clock time speed-up relative to SGD. We run SDD for 100k iterations to match the SGD baseline, SDD often converges earlier than that.

## 4.2 LARGE-SCALE THOMPSON SAMPLING

Next, we replicate the synthetic large-scale black-box function optimisation experiments of Lin et al. (2023), which consist of finding the maxima of functions mapping $[0,1]^8 \to \mathbb{R}$ sampled from Matérn-3/2 Gaussian process priors with 5 different length-scales and 10 random functions per

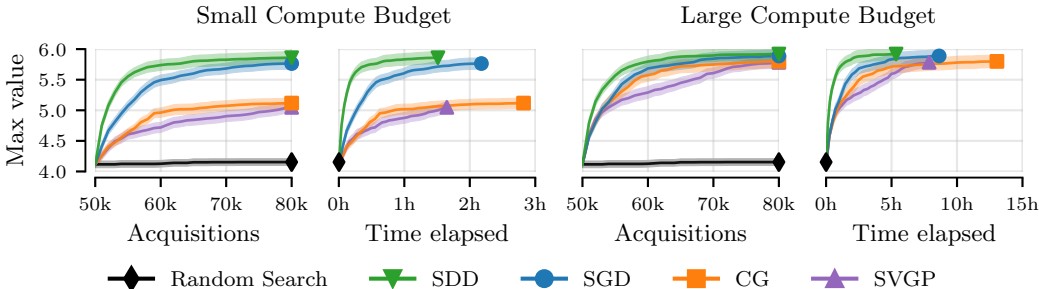

Figure 5: Results for the Thompson sampling task. Plots show mean and standard error of the maximum function values identified, across 5 length scales and 10 seeds, against both the number of observations acquired and the corresponding compute time on an A100 GPU. The compute time includes drawing posterior function samples and finding their maxima. All methods share an initial data set of 50k points, and take 30 steps of parallel Thompson sampling, acquiring 1k points at each.

length-scale, using parallel Thompson sampling (Hernández-Lobato et al., 2017). We set the kernel used by each model to match that of the unknown function. All methods are warm-started with the same 50k points chosen uniformly at random on the domain. We then run 30 iterations of the parallel Thompson sampling, acquiring 1000 points at each iteration. We include two variants of the experiment, one with a small compute budget, where SGD and SDD are run for 15k steps, SVGP is given 20k steps and CG is run for 10 steps, and one with a large budget, where all methods are run for 5 times as many steps. We present the results on this task in Figure 5, averaged over both length-scales and seeds, and a detailed breakdown in Appendix C. In both large and small compute settings, SDD makes the most progress, in terms of maximum value found, while using the least compute. The performance of SDD and SGD degrades gracefully when compute budget is limited.

## 4.3 MOLECULE-PROTEIN BINDING AFFINITY PREDICTION FOR DRUG DISCOVERY

In this final experiment, we show that Gaussian processes with SDD are competitive with graph neural networks in predicting binding affinity, a widely used filter in drug discovery (Pinzi and Rastelli, 2019; Yang et al., 2021). We use the DOCKSTRING regression benchmark of García-Ortegón et al. (2022), which contains five tasks, corresponding to five different proteins. The inputs are the graph structures of 250k candidate molecules, and the targets are real-valued affinity scores from the docking simulator *AutoDock Vina* (Trott and Olson, 2010). For each protein, we use a standard train-test splits of 210k and 40k molecules, respectively. We use Morgan fingerprints of dimension 1024 (Rogers and Hahn, 2010) to represent the molecules, and use a Gaussian process model based on the Tanimoto kernel of Ralaivola et al. (2005) with the hyperparameters of Tripp et al. (2023).

In Table 2, following García-Ortegón et al. (2022), we report $R^2$ values. Alongside results for SDD and SGD, we include results from García-Ortegón et al. (2022) for XGBoost, and for two graph neural networks, MPNN (Gilmer et al., 2017) and Attentive FP (Xiong et al., 2019), the latter of which is the state-of-the-art for this task. We also include the results for SVGP reported by Tripp et al. (2023). These results show that SDD matches the performance of Attentive FP on the ESR2 and FP2 proteins, and comes close on the others. To the best of our knowledge, this is the first time Gaussian processes have been shown to be competitive on a large-scale molecular prediction task.

## 5 CONCLUSION

We introduced stochastic dual descent, a specialised first-order stochastic optimisation algorithm for computing Gaussian process mean predictions and posterior samples. We showed that stochastic dual descent performs very competitively on standard regression benchmarks and on large-scale Bayesian optimisation, and matches the performance of state-of-the-art graph neural networks on a molecular binding affinity prediction task.

ACKNOWLEDGMENTS

JAL and SP were supported by the University of Cambridge Harding Distinguished Postgraduate Scholars Programme. AT was supported by Cornell University, jointly via the Center for Data Science for Enterprise and Society, the College of Engineering, and the Ann S. Bowers College of Computing and Information Science. JMHL acknowledges support from a Turing AI Fellowship under grant EP/V023756/1. CS gratefully acknowledges funding from the Canada CIFAR AI Chairs Program, Amii and NSERC.

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

## A CONVEX DUALITY AND UNIFORM APPROXIMATION BOUNDS

**Claim 1** (Strong duality). *We have that*

$$\min_{\alpha \in \mathbb{R}^n} L(\alpha) = -\lambda \min_{\alpha \in \mathbb{R}^n} L^*(\alpha),$$

*for $L, L^*$ defined per equations* (2) *and* (4) *respectively, with $\alpha^\star(b)$ minimising both sides.*

*Proof.* That $\alpha^\star(b)$ minimises both $L$ and $L^\star$ can be established from the first order optimality conditions. Now, for the duality, observe that we can write $\min_{\alpha \in \mathbb{R}^n} L(\alpha)$ equivalently as the constrained optimisation problem

$$\min_{u \in \mathbb{R}^n} \min_{\alpha \in \mathbb{R}^n} \frac{1}{2}\|u\|^2 + \frac{\lambda}{2}\|\alpha\|_K^2 \quad \text{subject to} \quad u = K\alpha - b.$$

Note that this is quadratic in both $u$ and $\alpha$. Introducing Lagrange multipliers $\beta \in \mathbb{R}^n$, in the form $\lambda\beta$, where we recall that $\lambda > 0$, the solution of the above is equal to that of

$$\min_{u \in \mathbb{R}^n} \min_{\alpha \in \mathbb{R}^n} \sup_{\beta \in \mathbb{R}^n} \frac{1}{2}\|u\|^2 + \frac{\lambda}{2}\|\alpha\|_K^2 + \lambda\beta^\mathsf{T}(b - K\alpha - u).$$

This is a finite-dimensional quadratic problem, and thus we have strong duality (see, e.g., Examples 5.2.4 in Boyd and Vandenberghe, 2004). We can therefore exchange the order of the minimum operators and the supremum, yielding the again equivalent problem

$$\sup_{\beta \in \mathbb{R}^n} \left\{ \min_{u \in \mathbb{R}^n} \frac{1}{2}\|u\|^2 - \lambda\beta^\mathsf{T}u \right\} + \left\{ \min_{\alpha \in \mathbb{R}^n} \frac{\lambda}{2}\|\alpha\|_K^2 - \lambda\beta^\mathsf{T}K\alpha \right\} + \lambda\beta^\mathsf{T}b.$$

Noting that the two inner minimisation problems are quadratic, we solve these analytically using the first order optimality conditions, that is $\alpha = \beta$ and $u = \lambda\beta$, to obtain that the above is equivalent to

$$\sup_{\beta \in \mathbb{R}^n} -\lambda\left( \frac{1}{2}\|\beta\|_{K+\lambda I}^2 - \beta^\mathsf{T}b \right) = -\lambda \min_{\beta \in \mathbb{R}^n} L^*(\beta).$$

The result follows by chaining the above equalities and relabelling $\beta \mapsto \alpha$. $\qquad\square$

To show equation (5), the uniform approximation bound, we first need to define reproducing kernel Hilbert spaces (RKHSes). Let $\mathcal{H}$ be a Hilbert space of functions $\mathcal{X} \to \mathbb{R}$ with inner product $\langle \cdot, \cdot \rangle$ and corresponding norm $\|\cdot\|_\mathcal{H}$. We say $\mathcal{H}$ is the RKHS associated with a bounded kernel $k$ if the reproducing property holds:

$$\forall x \in \mathcal{X}, \ \forall h \in \mathcal{H}, \qquad \langle k(x, \cdot), h \rangle = h(x).$$

That is, $k(x, \cdot)$ is the evaluation functional (at $x \in \mathcal{X}$) on $\mathcal{H}$. For observations $X$, let $\Phi \colon \mathcal{H} \to \mathbb{R}^n$ be the linear operator mapping $h \mapsto h(X)$, where $h(X) = (h(x_1), \ldots, h(x_n))$. We will write $\Phi^*$ for the adjoint of $\Phi$, and observe that $K$ is the matrix of the operator $\Phi\Phi^*$ with respect to a standard basis on $\mathbb{R}^n$ (that used implicitly throughout).

**Claim 2** (Uniform approximation). *For any $\alpha, \alpha' \in \mathbb{R}^n$,*

$$\|h_\alpha - h_{\alpha'}\|_\infty \leq \sqrt{\kappa}\|\alpha - \alpha'\|_K,$$

*where $\kappa = \sup_{x \in \mathcal{X}} k(x, x)$.*

*Proof.* First, observe that,

$$
\begin{aligned}
\|h_\alpha - h_{\alpha'}\|_\infty &= \sup_{x \in \mathcal{X}} |h_\alpha(x) - h_{\alpha'}(x)| && \text{(defn. of sup norm)} \\
&= \sup_{x \in \mathcal{X}} |\langle k(x, \cdot), h_\alpha - h_{\alpha'} \rangle| && \text{(reproducing property)} \\
&\leq \sup_{x \in \mathcal{X}} \|k(x, \cdot)\|_\mathcal{H} \|h_\alpha - h_{\alpha'}\|_\mathcal{H} && \text{(Cauchy-Schwarz)} \\
&\leq \sqrt{\kappa}\|h_\alpha - h_{\alpha'}\|_\mathcal{H}, && \text{(defn. of }\kappa\text{)}
\end{aligned}
$$

Now, observe that $h_\alpha = \Phi^*\alpha$ and $h_{\alpha'} = \Phi^*\alpha'$, and so we have the equalities

$$
\begin{aligned}
\|h_\alpha - h_{\alpha'}\|_\mathcal{H}^2 &= \langle \Phi^*(\alpha - \alpha'), \Phi^*(\alpha - \alpha') \rangle && \text{(defn. norm)} \\
&= \langle \alpha - \alpha', \Phi\Phi^*(\alpha - \alpha') \rangle && \text{(defn. adjoint)} \\
&= \|\alpha - \alpha'\|_K^2, && \text{(defn. }K\text{)}
\end{aligned}
$$

where for the final equality, we note that $K$ is the matrix of the operator $\Phi\Phi^*$ with respect to the standard basis. Combining the above two displays yields the claim. $\qquad\square$

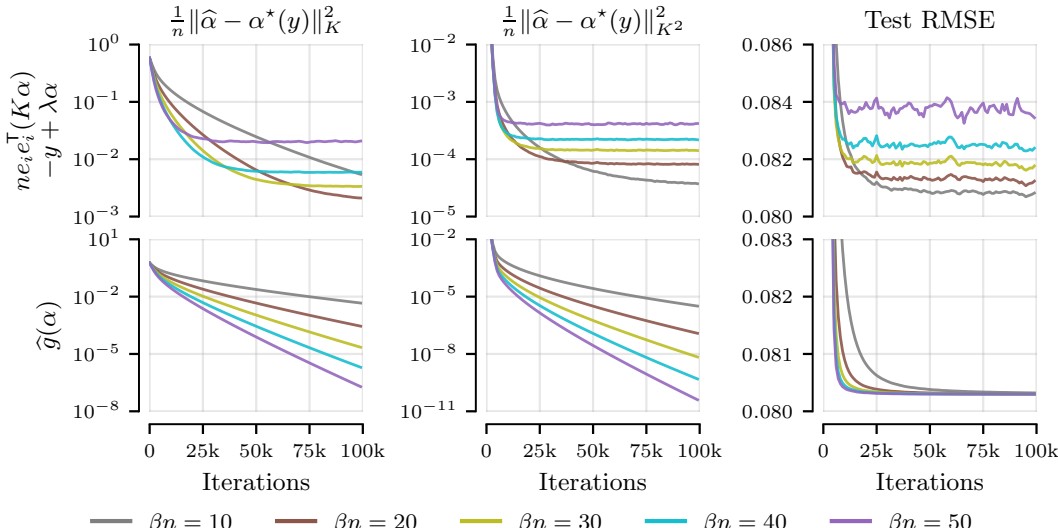

Figure 6: Comparison of dual stochastic gradient descent on the POL with a momentum parameter $\rho = 0.9$, averaging parameter $r = 0.001$, batch size $B = 512$, and varying step-sizes $\beta n$. For the mixed additive-multiplicative noise gradient estimator in the top row, higher step-size leads to faster convergence, but worse asymptotic result. For our recommended multiplicative noise estimator in the bottom row, higher step-size improves both the speed of convergence and the asymptotic result.

## B    EFFECTS OF VARYING STEP-SIZE AND BATCH-SIZE

In Figures 6 to 8 we examine the trade-offs related to step-size and batch-size for stochastic dual gradient descent with gradient estimators either of the mixed additive-multiplicative type, used by SGD, or the purely multiplicative type, recommended in this work. In short, trade-offs exist in the additive-multiplicative case, and we cannot make clear recommendations. For purely multiplicative noise, the picture is clearer:

- Step-size should be chosen as large as possible while avoiding divergence, to improve both the rate of convergence and the asymptotic quality of the result.
- Batch-size should be chosen as small as possible while avoiding divergence, since a larger batch incurs a larger per-step computational cost without affecting the convergence of the algorithm.

Of course, this then leaves a step-size versus batch-size trade-off for SDD, translating into a trade-off between the rate of convergence and the asymptotic behaviour versus wall-clock time. This trade-off will need to be addressed on a case-by-case basis.

## C    ADDITIONAL DETAILS ON EXPERIMENTAL SETUPS AND RESULTS

We use the implementation and hyperparameter configurations from Lin et al. (2023) for the SGD, SVGP and CG baselines,[1] which uses the `jax` library (Bradbury et al., 2018). Some more detail on the implementations is as follows.

**CG**  We implement CG using the `jax.scipy` library, with a pivoted Cholesky preconditioner of size 100 computed using the `TensorFlow Probability` library.

**SGD**  We use minibatches of 512 to estimate the fit term of equation (6), and 100 random features for the regulariser. For UCI and Bayesian optimisation experiments, random Fourier features are used. For the molecule task, random hashes, as described in Tripp et al. (2023). SGD uses

---

[1]HTTPS://GITHUB.COM/CAMBRIDGE-MLG/SGD-GP

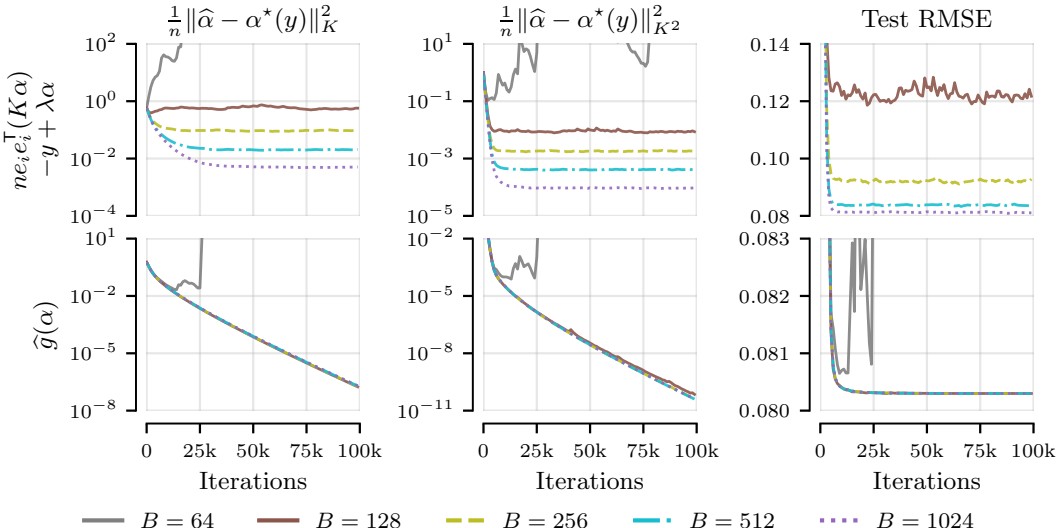

Figure 7: Dual stochastic gradient descent on the POL with a momentum parameter $\rho = 0.9$, averaging parameter $r = 0.001$, step-size $\beta n = 50$, and varying batch sizes $B$, with metrics plotted against the number of iterations. For the mixed additive-multiplicative noise gradient estimator in the top row, higher batch size improves the final performance. For our preferred multiplicative noise estimator in the bottom row, batch sizes has little effect on performance, so long as it is not so low that it causes the optimisation to diverge.

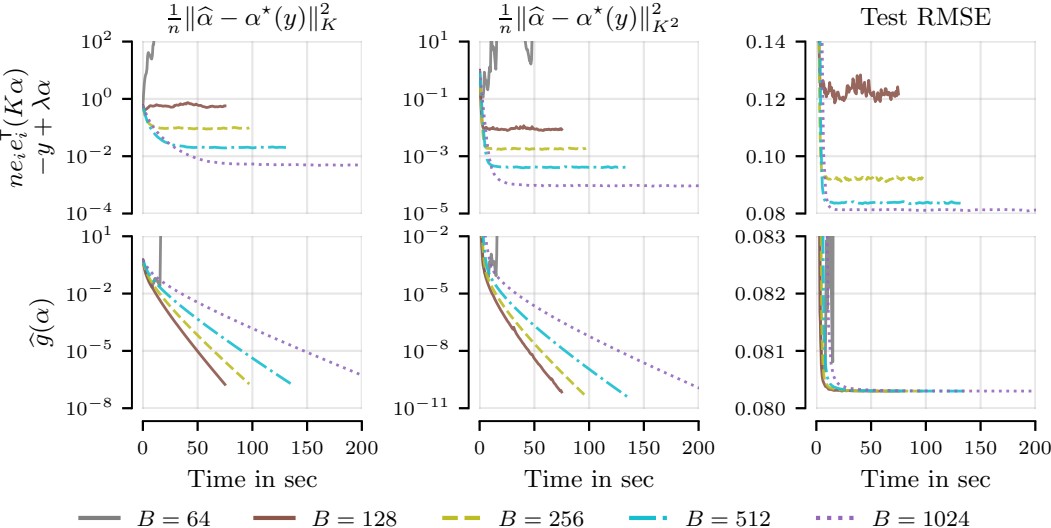

Figure 8: Dual stochastic gradient descent on the POL with a momentum parameter $\rho = 0.9$, averaging parameter $r = 0.001$, step-size $\beta n = 50$, and varying batch sizes $B$, with metrics plotted against wall-clock time. For the mixed additive-multiplicative noise gradient estimator, conclusions match those in Figure 7. For our preferred multiplicative noise estimator in the bottom row, since performance for batch sizes that lead to convergence is similar, performance is best for the smallest batch-size that does not lead to divergence, as batch-size increases the per-step computation cost.

Nesterov's momentum of $0.9$ and geometric iterate averaging, as implemented in `optax`, with $r = 100/T$, where $T$ is the total number of SGD iterations. We clip the 2-norm of the gradient estimates to $0.1$. These settings are replicated from Lin et al. (2023), which in turn takes these from Antorán et al. (2023).

**SVGP** We initialise inducing points using `k-means` and `k-means++`. We then optimise all variational parameters, including inducing point locations, by maximising the ELBO with the Adam optimiser until convergence, using the `GPJax` library (Pinder and Dodd, 2022).

## C.1 UCI Regression

For all methods, we use a Matérn-$3/2$ kernel with a fixed set of hyperparameters chosen by Lin et al. (2023) via maximum likelihood. To ease reproducibility, we make the full hyperparameter set available as a Python file in our source code HERE. SGD uses $\beta n = 0.5$ to estimate the mean function weights, and $\beta n = 0.1$ to draw samples. For SDD, we use step-sizes $\beta n$ which are 100 times larger, except for ELEVATORS, KEGGDIR and BUZZ, where this causes divergence; there, we use a $10\times$ larger step-size instead. SDD and SGD are both run for 100k steps with batch size $B = 512$ for both the mean function and posterior samples. For CG, we use a maximum of 1000 steps for data sets with $N \leq 50$k, and a tolerance of $0.01$. On the four largest data sets, the per step cost of CG is too large to run 1000 steps, and we run 100 steps instead. For SVGP, the number of inducing points is chosen such that the compute cost approximately matches that of other methods: 3000 for the smaller five data sets and 9000 for the larger four. Negative log-likelihood computations are done by estimating the predictive variances using 64 posterior samples, with 2000 random Fourier features used to approximate the requisite prior samples.

## C.2 Large-scale Thompson Sampling

We draw target functions $\mathcal{X} \to \mathbb{R}$ from a Gaussian process prior with a Matérn-$3/2$ kernel and length scales $(0.1, 0.2, 0.3, 0.4, 0.5)$, using 2000 random Fourier features. For each length scale, we repeat the experiment for 10 seeds. All methods use the same kernel that was used to generate the data.

We optimise the target functions on $\mathcal{X} = [0, 1]^8$ using parallel Thompson sampling (Hernández-Lobato et al., 2017). That is, we choose $x_{\text{new}} = \arg\max_{x \in \mathcal{X}} f_n$ for a set of posterior function samples drawn in parallel. We replicate the multi-start gradient optimisation maximisation strategy of Lin et al. (2023) (see therein). For each function sample maximum, we evaluate $y_{\text{new}} = g(x_{\text{new}}) + \varepsilon$ with $\varepsilon$ drawn from a zero-mean Gaussian with variance of $10^{-6}$. We then add the pair $(x_{\text{new}}, y_{\text{new}})$ to the training data. We use an acquisition batch size of 1000. We initialise all methods with a data set of 50k observations sampled uniformly at random from $\mathcal{X}$.

Here, SGD uses a step-size of $\beta n = 0.3$ for the mean and $\beta n = 0.0003$ for the samples. SDD uses step-sizes that are $10\times$ larger: $\beta n = 3$ for the mean and $\beta n = 0.003$ for the samples.

## C.3 Molecule-protein Binding Affinity Prediction

We use the data set and standard train-test splits from García-Ortegón et al. (2022), which were produced by structure-based clustering to avoid similar molecules from occurring both in the train and test set. We perform all the preprocessing steps for this benchmark outlined by García-Ortegón et al. (2022), including limiting the maximum docking score to 5.

**Fingerprints, Tanimoto Kernel and Random Features** Molecular fingerprints are a way to encode the structure of molecules by indexing sets of subgraphs present in a molecule. There are many types of fingerprints. Morgan fingerprints represent the subgraphs up to a certain radius around each atom in a molecule (Rogers and Hahn, 2010). The fingerprint can be interpreted as a sparse vector of counts, analogous to a 'bag of words' representation of a document. Accordingly, the Tanimoto coefficient $T(x, x')$, also called the Jaccard index, is a way to measure similarity between fingerprints, given by

$$T(x, x') = \frac{\sum_i \min(x_i, x_i')}{\sum_i \max(x_i, x_i')}.$$

This function is a valid kernel and has a known random feature expansion using random hashes (Tripp et al., 2023). The feature expansion builds upon prior work for fast retrieval of documents

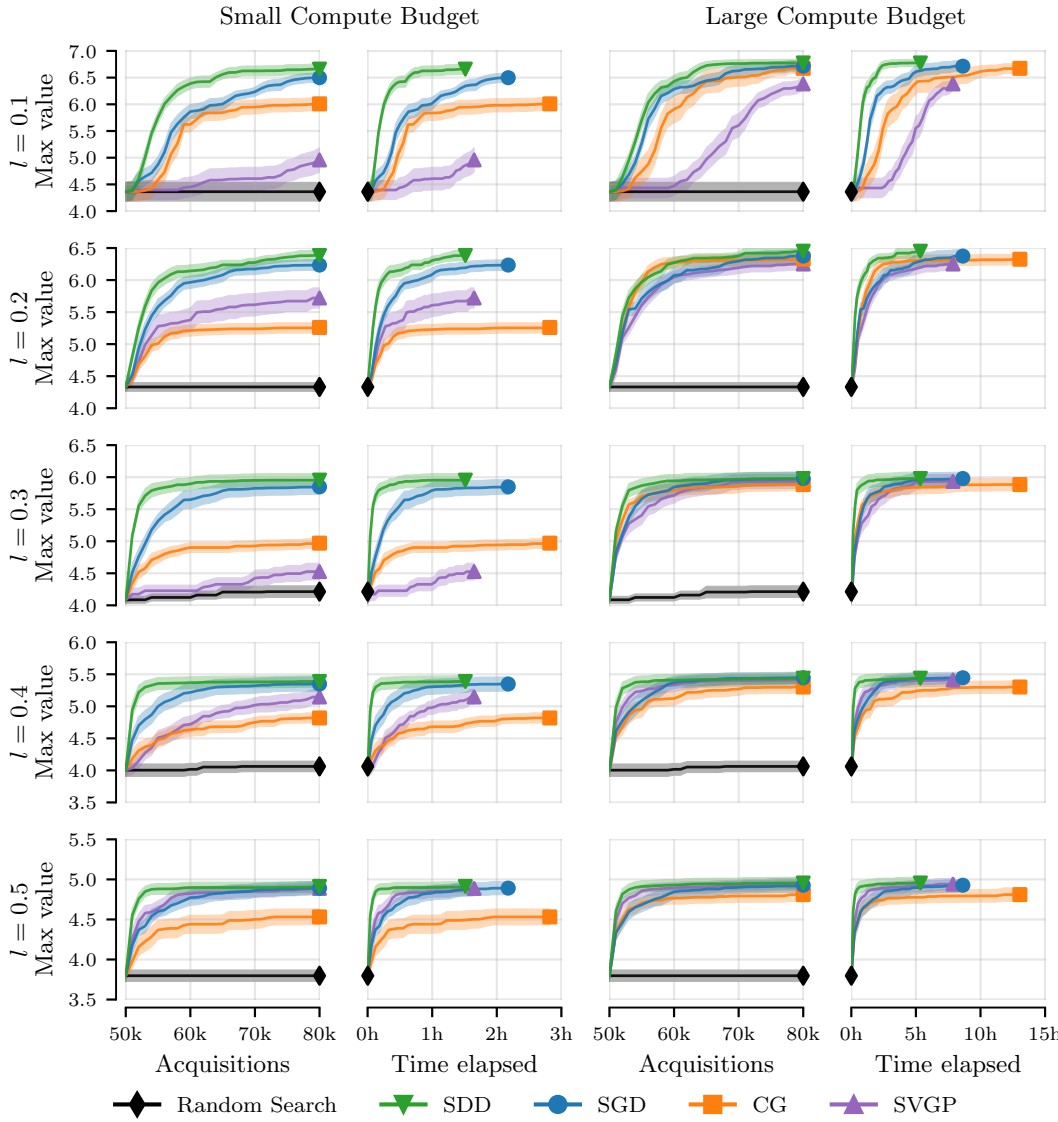

Figure 9: Maximum function values, with mean and standard error across 10 seeds, obtained by parallel Thompson sampling, for functions with different length-scales $l$, plotted as functions of acquisition steps and the compute time on an A100 GPU. All methods share an initial data set of 50k points, and take 30 Thompson steps, acquiring a batch of 1000 points in each. The algorithms perform differently across the length-scales: CG performs better in settings with smaller length-scales, which give better conditioning; SVGP tends to perform better in settings with larger length-scales and thus higher smoothness; SGD and SDD perform well in both settings.

using random hashes that approximate the Tanimoto coefficient; that is, a distribution $P_h$ over hash functions $h$ such that

$$P_h(h(x) = h(x')) = T(x, x').$$

Per Tripp et al. (2023), we extend such hashes into random *features* by using them to index a random tensor whose entries are independent Rademacher random variables, and use the random hash of Ioffe (2010).

**Gaussian Process Setup**   The results for the SVGP baseline are taken from Tripp et al. (2023). As the Tanimoto kernel itself has no hyperparameters, the only kernel hyperparameters are a constant scaling factor $A > 0$ for the kernel and the noise variance $\lambda$, and a constant GP prior mean $\mu_0$ (the Gaussian process regresses on $y - \mu_0$ in place of $y$). These are chosen by using an exact GP to a randomly chosen subset of the data and held constant during the optimisation of the inducing points. The values of these are given in Table 3. The same values are also used for SGD and SDD to ensure that the differences in accuracy are solely due to the quality of the GP posterior approximation. The SGD method uses 100-dimensional random features for the regulariser.

Table 3: Hyperparameters for all Gaussian process methods used in the molecule-protein binding affinity experiments of Section 4.3.

| Data | ESR2 | F2 | KIT | PARP1 | PGR |
|---|---|---|---|---|---|
| $A$ | 0.497 | 0.385 | 0.679 | 0.560 | 0.630 |
| $\lambda$ | 0.373 | 0.049 | 0.112 | 0.024 | 0.332 |
| $\mu_0$ | -6.79 | -6.33 | -6.39 | -6.95 | -7.08 |

