# OpenReview forum: "Stochastic Gradient Descent for Gaussian Processes Done Right"
_ICLR.cc/2024/Conference — ICLR 2024 poster_

### Official Review · Reviewer_NBgz · 2023-10-20

**Soundness:** 4 excellent
**Presentation:** 4 excellent
**Contribution:** 4 excellent
**Rating:** 8
**Confidence:** 4

**Summary:**

The authors consider the problem of computing a Gaussian process posterior, specifically its mean and random draws from it. While the naive computation scales cubically in the number of observations, the authors propose a iterative solver with linear cost per iteration. The idea behind this solver is that the expensive quantity in the GP posterior (kernel matrix inverse) can be thought of as a minimiser of a particular regression problem, which can be solved iteratively with gradient-based methods. The authors consider two formulations of such a regression problem (primal and dual), study their convergence properties, as well as discuss randomised gradients computations to achieve linear computational cost. The proposed algorithm is shown to perform competitively on a number of benchmarks.

**Strengths:**

+ The paper is clearly written and is easy to follow
+ The differences to the closely related work of Liu et al. (2023) are clearly discussed
+ I think the results are quite significant for the community. I was especially interested to see that the proposed algorithm performs competitively in comparison to a neural network in Table 2.

**Weaknesses:**

I didn't notice any significant weaknesses.

**Questions:**

- In Fig. 1 you note that the primal gradient makes more progress in K^2-norm while the dual one in K-norm (with the same step size). However, in the left panel of Fig. 1 it seems that for a few iterations in the beginning of optimisation, the primal gradient was also making more progress than dual in the K-norm. Why do you think it is the case?

- The GP hyper-parameters (e.g. observational noise variance, kernel parameters, etc.) are typically estimated by maximising the marginal log-likelihood using gradient-based methods. Do you think it could be possible run the gradient-based hyper-parameters inference jointly with the posterior inference that you discussed in this paper?

---

> ### Author Response · Authors · 2023-11-16
> **Thank you for your review, please see our responses to your questions!**
>
> Thank you very much for your review! We are delighted to hear that our results are significant—thank you very much for this comment. Let us address your questions:
>
> ---
>
> **1. Early behaviour in optimisation iterations**
>
> > In Fig. 1 you note that the primal gradient makes more progress in K^2-norm while the dual one in K-norm... in the left panel of Fig. 1 it seems that for a few iterations in the beginning of optimisation, the primal gradient was also making more progress than dual in the K-norm.
>
> Regarding Figure 1, specifically the ordering of performance right at the beginning of optimisation: we’ve thought about this, and we genuinely do not know! We think that this behaviour is relatively rare, as it occurs way out of convergence, with relatively few iterations spent in this regime, which should make it unlikely to affect behaviour in practice very much.
>
> **2. Kernel hyperparameter optimisation / Bayesian model selection**
> > The GP hyper-parameters ... are typically estimated by maximising the marginal log-likelihood using gradient-based methods. Do you think it could be possible run the gradient-based hyper-parameters inference jointly with the posterior inference that you discussed in this paper?
>
> Our method can indeed be easily combined with standard approaches to kernel hyperparameter optimisation. **We have added a demonstration of this in Appendix C.7** (we use an approach similar to [Gardner et al. 2018](https://arxiv.org/pdf/1809.11165.pdf)), and include some empirical results to that end where **we match performance of marginal likelihood optimisation with exact linear system solves (see the figure [here](https://anonymous.4open.science/r/SDD-GPs-5486/rebuttal_plots/mll_optim.pdf)).** Thank you for pointing out that this may be interesting to our readers to highlight!

---

> > ### Comment · Reviewer_NBgz · 2023-11-20
> >
> > Thank you for your reply! I will read the interactions with other reviewers and get back to you if I have more questions later.

---

### Official Review · Reviewer_1x4h · 2023-10-31

**Soundness:** 3 good
**Presentation:** 3 good
**Contribution:** 2 fair
**Rating:** 6
**Confidence:** 4

**Summary:**

This paper introduced a stochastic dual gradient descent algorithm for kernel ridge regression and sampling. The stochastic dual descent algorithm admits better-conditioned gradients and a faster convergence rate compared to the SGD proposed by Lin et al. (2023). With the selected kernels, experimental results showed competitive performance with a number of SOTA methods on UCI regression/Bayesian optimization/ molecular binding affinity prediction tasks. Overall, the paper is easy to follow and well-written, while technical contributions seem to be below the bar of ICLR.

**Strengths:**

The strengths are:
(1) Some fresh insights from the optimization and kernel communities were explored.
(2) Uniform approximation bound and duality of objectives were both analyzed.
(3) Different randomized gradients and convergence performance were compared.

**Weaknesses:**

Some suggestions on improving the weakness points are:
(1) More figures/tables to explicitly show the weakness/instability of the baseline methods are expected.
(2) Sharing more insights into the algorithm settings, such as the choice of geometric averaging, the effect/influence on the sparsity of the unbiased estimator \hat(g)(\alpha), etc, are expected.
(3) A theoretical convergence analysis is expected (not only some figures).

**Questions:**

1. In Figure 1, we can not see the primal gradient descent becomes unstable and diverges for $\beta n$>0.1. Please show the unstable or compare the evaluated conditional numbers. Under higher step sizes, why does the gradient descent of the primal return $NaN$ (any possible reasons)?
2. Figure 2 shows the random coordinate estimate with a step size equal to 50. what is the performance on varied step sizes? Can any explanation of the rising part (the blue dashed line) in the middle figure in Figure 2 be given?
3. What is the step size used to generate the Figure 3? It seems less than 50 and has a competitive rate compared to the random feature estimate shown in Figure 2. Extra clarification and comparison would be better.
4. How do different batch sizes affect the overall convergence?
5. It is better to add a test where samples are generated by a non-stationary kernel, to show the ability of the random coordinate estimate. (to distinguish with the random Fourier features)
6. what is the difference between the $\beta n$ in the main texts and the $\beta$ in Algorithm 1?
7. The green dashed line is missing in Figure 3.

---

> ### Author Response · Authors · 2023-11-16
> **Thank you for your feedback, please consider our responses (1/3)**
>
> Thank you for your thorough review. We are glad you think the paper is easy to follow and well-written.
>
> First of all we would like to address the overall comment regarding our technical contributions.
> > ...technical contributions seem to be below the bar of ICLR
>
>
> All applied kernel/GP work we are familiar with, which uses SGD methods, makes suboptimal algorithmic design choices (e.g. [Lin et al. 2023](https://arxiv.org/pdf/2306.11589.pdf) (NeurIPS2023) and [Dai et. al. 2014](https://arxiv.org/pdf/1407.5599.pdf) (NeurIPS2014)). On this basis, we believe that **a simple algorithm that combines recent theoretical advancements in a principled way, and is the first to reliably outperform conjugate gradient-based algorithms used in state-of-the-art GP software packages, can be of value to the ICLR community.** We consider the simplicity of our approach a strength!
>
>
> We go on to address your remaining concerns and questions.
>
> -----
>
>
> **1. Instability and divergence of baselines.**
> > More figures/tables to explicitly show the weakness/instability of the baseline methods are expected.
>
> **We have added a number of further comparisons where we show failure modes of the baselines we consider**, namely SGD from [Lin et al. 2023](https://arxiv.org/pdf/2306.11589.pdf), CG, and SVGP.
>
> 1. In Figures 12 and 13 in Appendix C.6, also available [here](https://anonymous.4open.science/r/SDD-GPs-5486/rebuttal_plots/primal_lr_divergence_pol_10.pdf) and [here](https://anonymous.4open.science/r/SDD-GPs-5486/rebuttal_plots/primal_lr_divergence_pol_20.pdf), you can see how the primal SGD baseline diverges very quickly at even moderately high learning rates.
> 2. In Figure 6 in Appendix C.1 ([here](https://anonymous.4open.science/r/SDD-GPs-5486/rebuttal_plots/toy_comparison_rebuttal.pdf)), you can see failure modes for CG (on the infill asymptotics task) and SVGP (large-domain asymptotics task). Our dual SGD method performs robustly in both tasks.
> 3. In Figure 11 in Appendix C.5 ([here](https://anonymous.4open.science/r/SDD-GPs-5486/rebuttal_plots/ill_conditioned_pol.pdf)), you can see that CG struggles in ill-conditioned settings, where the dual SGD method is quite robust.
>
> **2. Algorithmic insights**
> > Sharing more insights into the algorithm settings, such as the choice of geometric averaging, the effect/influence on the sparsity of the unbiased estimator \hat(g)(\alpha), etc, are expected.
>
> Sparsity of the random coordinate estimate plays no role in our recommendation of this gradient estimate. Rather we point it out to help the reader understand what this estimate looks like.
>
> Regarding averaging, **our main point is that iterate averaging may not be necessary for convergence under multiplicative noise (Section 3.3)**, and can be completely done away with. Indeed, arithmetic iterate averaging slows down convergence empirically (see Figure 3).
>
> However, we find that geometric averaging, which is much softer a form of averaging (places much more weight on recent iterates), is useful in practice. We thus recommend its use.
>
> **3. Theoretical analysis**
> > A theoretical convergence analysis is expected (not only some figures).
>
>
>
> Our paper is targeted at GP researchers and practitioners. While using SGD for GP inference has recently been found to be very promising ([Lin et al. 2023](https://arxiv.org/pdf/2306.11589.pdf) and [Dai et. al. 2014](https://arxiv.org/pdf/1407.5599.pdf)), **previous work makes a number of suboptimal choices when it comes to the details of the algorithm's formulation. Our contribution is handling these details carefully, highlighting to researchers and practitioners how to make SGD inference even more effective**, particularly on harder problems like the graph kernel comparison.
>
>
> In the **full-batch (non stochastic) case, the analysis of the algorithm is standard** ([Nesterov 1983](https://www.mathnet.ru/php/archive.phtml?wshow=paper&jrnid=dan&paperid=46009&option_lang=eng)), and it is known that the dual objective will result in faster convergence than its primal counterpart due to conditioning. See, for instance Section 3.7, page 289, of [Bubeck 2015](https://arxiv.org/pdf/1405.4980.pdf), a textbook on convex optimisation.
>
> Our full algorithm makes use of an estimator with multiplicative noise. **The convergence of Stochastic GD under multiplicative noise is currently at the forefront of theoretical research and beyond the scope of our more practical work.** We have extended our discussion of pertinent work in this space in Section 3.4. We hope that our work's demonstration of the empirical performance achievable with SGD under multiplicative noise will motivate research in this area by the optimisation community.
>
> (continued below)

---

> > ### Comment · Reviewer_1x4h · 2023-11-21
> > **Many thanks for your reply**
> >
> > I appreciate the detailed reply which solves most of my questions and concerns. I will raise my score due to the great effort the authors put into this work. The only minor thing I want to point out is perhaps the authors shouldn't disclose their names (through Arxiv) in the review/rebuttal phase.

---

> ### Author Response · Authors · 2023-11-16
> **Thank you for your feedback, please consider our responses (2/3)**
>
> **4. SGD baseline divergence at large step-sizes**
> > In Figure 1, we can not see the primal gradient descent becomes unstable and diverges for >0.1. Please show the unstable or compare the evaluated conditional numbers. Under higher step sizes, why does the gradient descent of the primal return nan (any possible reasons)?
>
> **We include the requested figures showing the divergence of primal SGD with high step-sizes, [here](https://anonymous.4open.science/r/SDD-GPs-5486/rebuttal_plots/primal_lr_divergence_pol_10.pdf) and [here](https://anonymous.4open.science/r/SDD-GPs-5486/rebuttal_plots/primal_lr_divergence_pol_20.pdf), and Appendix C.6. This behaviour quickly exhausts the numerical floating point precision, leading to  overflows and thus NaNs.**
>
> We did not include these in the original text because the loss curves diverge so quickly that they cannot be viewed simultaneusly with the optimisation trajectories of the non-diverging methods. We run our algorithm for 100k iterations. After only 10 iterations, amplifying oscillations are visible and after 20 iterations an exponential trend is already present.
>
> **5. Varying step-sizes**
> > Figure 2 shows the random coordinate estimate with a step size equal to 50. what is the performance on varied step sizes?
>
>
> We have conducted **further experiments** under the same settings with **varying step-sizes** for both random coordinate estimators (the one with additive and the one with multiplicative noise). **Please see the corresponding figure [here](https://anonymous.4open.science/r/SDD-GPs-5486/rebuttal_plots/lr_comparison_pol.pdf)** and more details in Appendix C.3.
>
>
> The performance of the additive noise estimator plateaus earlier as the step-size increases. Smaller step-sizes converge slower but eventually reach better solutions, such that a trade-off is necessary: the step-size is proportional to the speed of convergence but inversely proportional to the final performance. In contrast, the multiplicative noise estimator consistently improves both convergence speed and final performance as the step-size is increased, as long as the step-size is not large enough to cause divergence.
>
>
>
>
> **6. Step size in Figure 3**
> > What is the step size used to generate the Figure 3?
>
> The step-size used to generate Figure 3 was **$\beta n = 50$. We added this to the figure caption now**.
>
> **7. Random feature plot line not matching in Figures 2,3 + rising blue line**
> > It seems less than 50 and has a competitive rate compared to the random feature estimate shown in Figure 2. Extra clarification and comparison would be better..
>
>  > Can any explanation of the rising part (the blue dashed line) in the middle figure in Figure 2 be given?
>
> Thank you for bringing up this source of confusion. Only the dashed light blue line should be nearly identical in both cases. **Both plots use a step-size of 50**, but different batch sizes. However, we have discovered the **lines are different due to the floating point precision** used when running the experiment. Figure 2 was generated with 32-bit floating point, which becomes unstable when the error is very small (~ 1e-7). We have now replaced this with 64-bit floating point, so **this is now fixed. Please see the fixed plot [here](https://anonymous.4open.science/r/SDD-GPs-5486/rebuttal_plots/batch_variants_pol.pdf). To improve clarity, we also added "random features" and "random coordinates" to the labels in Figure 2.**
>
> (continued below)

---

> ### Author Response · Authors · 2023-11-16
> **Thank you for your feedback, please consider our responses (3/3)**
>
> **8. Effect of batch size**
> > How do different batch sizes affect the overall convergence?
>
> **We conducted additional experiments with varying batch sizes. Please see our figures [here](https://anonymous.4open.science/r/SDD-GPs-5486/rebuttal_plots/batch_size_comparison_pol.pdf) and [here](https://anonymous.4open.science/r/SDD-GPs-5486/rebuttal_plots/batch_size_comparison_pol_per_time.pdf)** and Appendix C.4 for further details.
>
> Smaller batch sizes degrade the final performance of the additive noise estimator because the amount of additive noise is inversely proportional to the batch size. In contrast, the multiplicative noise estimator produces nearly identical optimisation traces with different batch sizes, as long as the batch size is not small enough to cause divergence. Therefore, increasing the batch size consistently improves the final performance of the additive noise estimator while it does not impact the performance of the multiplicative noise estimator, given that the algorithm converges.
>
> In terms of performance vs time, a larger batch size requires additional compute per step. Thus, the additive noise estimator performance is proportional to the batch size and the multiplicative noise estimator performance per time is inversely proportional to the batch size, as long as the algorithm converges. Therefore, the multiplicative noise estimator should use the smallest batch size which does not cause divergence. Selecting such a batch size a priori is, in general, non-trivial. We have included this discussion in Appendix C4.
>
>
> **9. Non-stationary kernels**
> > It is better to add a test where samples are generated by a non-stationary kernel, to show the ability of the random coordinate estimate. (to distinguish with the random Fourier features)
>
> Thank you for the question. **Our optimisation method does not require the existence of closed-form random features: for computing the posterior mean, we can work with any kernel**. We only introduce feature-based gradient estimators due to their use in previous literature ([Lin et al. 2023](https://arxiv.org/pdf/2306.11589.pdf) and [Dai et. al. 2014](https://arxiv.org/pdf/1407.5599.pdf)). We ultimately recommend against their use for gradient estimation due to producing additive rather than multiplicative noise.
>
> We do use random features for efficiently sampling from the GP prior, which is needed, for instance, in the Bayesian optimisation experiments. We similarly use random hashing for the Tanimoto kernel in the graph experiment. In non-stationary kernels, efficient sampling can often be done on a case-by-case basis, but we prefer not to focus on it as it is orthogonal to our contributions.
>
> **10. $\beta$ vs. $\beta n$**
> > what is the difference between the $\beta$ in the main texts and the $\beta n$ in Algorithm 1?
>
> The $\beta n = \beta \times n$ is simply the learning rate $\beta$ times $n$, the size of the training data set. This formulation allows us to **keep the step-size $\beta$ times the data set size $n$ constant across data sets**. We mostly use $\beta n$ throughout our writing, since the algorithm ultimately depends on the respective product.
>
>
> **11. Figure 3 green dashed line**
>
> > The green dashed line is missing in Figure 3.
>
> Thank you for pointing this out! **We have updated the caption to reflect that this refers to the olive green line.**
>
> ----
>
> **Summarising:** thank you again for your feedback, which has made our experimental evaluation much more comprehensive. Please let us know what you think of the updated results!

---

### Official Review · Reviewer_eaDf · 2023-10-31

**Soundness:** 3 good
**Presentation:** 3 good
**Contribution:** 3 good
**Rating:** 8
**Confidence:** 3

**Summary:**

This paper uses insights drawn from the application of gradient descent in the kernel and optimisation communities to develop a stochastic gradient descent approach for Gaussian processes. In particular this method is useful in regresssion to approximate the posterior mean and to draw samples from the GP posterior. This method, stochastic dual descent, is compared to conjugate gradient, stochastic gradient descent and stochastic variational Gaussian processes on regression benchmarks and molecular binding affinity
prediction.

**Strengths:**

This is a well written paper that considers an interesting problem. The use of several benchmarks in the experimental section and comparison with recent work is a plus.

The justification for use of the dual objective as well as the illustrative example is clear.

The reason behind the choice of random coordinate estimates is well done.

**Weaknesses:**

It would be useful to emphasise that this work is useful when the Kernel is already known. Comments on whether these methods would be useful in hyperparameter estimation would be useful.

The claim that the method can be implemented in a few lines of code should be demonstrated. The repo given does not clearly illustrate this using a simple example.

The paper would benefit from a visualisation comparing samples from a GP using SDD to an exact GP fit to show that the samples lie within the confidence interval.

**Questions:**

What are the implications of limiting the kernel to the form $\sum_{j=1}^mz_jz_j^T$?

How does ill conditioning affect the performance of the method?

---

> ### Author Response · Authors · 2023-11-16
> **Thank you for your review, please consider our response below!**
>
> Thank you very much for your time reviewing our paper! We are very happy to hear our work was well-written, and focused on an interesting problem. Let us address the key questions:
>
> ----
> **1. Kernel and hyperparameter estimation**.
>
> > It would be useful to emphasise that this work is useful when the Kernel is already known.
>
> Thank you for pointing this out: **We have further clarified this in our introduction** with the language "As a result of the careful choices behind our method, we significantly improve upon the results of [Lin et al. 2023](https://arxiv.org/pdf/2306.11589.pdf) for stochastic gradient descent **in the fixed kernel setting.".**
>
> Our work solely focuses on improving SGD inference and thus employs the setting of [Lin et al.](https://arxiv.org/pdf/2306.11589.pdf), where the kernel is fixed.
>
> >  Comments on whether these methods would be useful in hyperparameter estimation would be useful.
>
> Our method can indeed be easily combined with standard approaches to kernel hyperparameter optimisation using the usual trick of applying the Hutchinson trace estimator on the marginal likelihood's gradient (see for instance [Gardner et al. 2018](https://arxiv.org/pdf/1809.11165.pdf)). As a proof of concept, **we have added a demonstration of this to Appendix C.7, and include some empirical results to that end where we match performance to standard optimisation of the marginal likelihood (see [here](https://anonymous.4open.science/r/SDD-GPs-5486/rebuttal_plots/mll_optim.pdf)).**
>
> **2. Ease of implementation**
>
> > The claim that the method can be implemented in a few lines of code should be demonstrated.
>
> Thank you for your suggestion! **We support our claim that SDD is easy to implement, by including a very simple [Jupyter Notebook](https://anonymous.4open.science/r/SDD-GPs-5486/sdd.ipynb)** that implements our full method in NumPy from scratch and runs it on a toy problem. Without counting data set generation and kernel definition, a basic version of our algorithm can be implemented with only **20 lines of code**.
>
> **3. Visualisation of samples**
>
> > The paper would benefit from a visualisation comparing samples from a GP using SDD to an exact GP fit to show that the samples lie within the confidence interval.
>
> We agree this would strengthen the paper! **We have added a visual comparison to exact GP, and to our CG and SVGP baselines, on two toy examples considered in Lin et al. 2023. See the figure [here](https://anonymous.4open.science/r/SDD-GPs-5486/rebuttal_plots/toy_comparison_rebuttal.pdf) and Appendix C.1.**
>
> **4. Kernel of the form $\sum z_j z_j^T$**
>
> > What are the implications of limiting the kernel to the form $\sum_{j=1}^m z_j z_j^T$?
>
> Thank you for drawing us to this potential misunderstanding. We wish to clarify that **our method does not require the kernel to be finite-dimensional.** We only mention this family of approaches in the context of random feature gradient estimators. We study these due to their use in previous literature ([Lin et al. 2023](https://arxiv.org/pdf/2306.11589.pdf) and [Dai et. al. 2014](https://arxiv.org/pdf/1407.5599.pdf)), but ultimately recommend against their use due to producing additive rather than multiplicative noise. To avoid confusion, we have **added a line in Section 3.2**: "For the sake of exposition (and exposition only, this is not an assumption of our algorithm), we assume that we are in a $m$-dimensional (finite) linear model setting,...".
>
> **5. Effect of ill-conditioning**
>
> > How does ill conditioning affect the performance of the method?
>
> Thank you for your question!
> In general, methods to solve linear systems are very dependent on the respective condition numbers. However, for Gaussian processes, [Lin et al. 2023](https://arxiv.org/pdf/2306.11589.pdf) show that SGD is significantly less sensitive to conditioning than CG, which can take an arbitrary amount of time to converge (and obtain good results) for sufficiently ill-conditioned systems.
>
> For SGD, **conditioning determines the maximum step-size which can be used without diverging.** Since the dual gradient presents better conditioning, as it depends on the kernel matrix eigenvalues as opposed to their squares, we can use much larger step-sizes. This can be seen in Figure 1 in the main paper, and in a new figure which we added to the appendix [here](https://anonymous.4open.science/r/SDD-GPs-5486/rebuttal_plots/lr_comparison_pol.pdf).
>
> We further demonstrate this in Appendix C.5 where we compare both methods on the pol data set while setting the noise standard deviation to 0.001. The figure is also shown [here](https://anonymous.4open.science/r/SDD-GPs-5486/rebuttal_plots/ill_conditioned_pol.pdf).

---

> > ### Comment · Reviewer_eaDf · 2023-11-23
> >
> > The authors have addressed all my concerns in a thorough manner. In addition to the work addressing the concerns of other reviewers, I feel that the paper is greatly improved. I have adjusted my score upwards.

---

### Official Review · Reviewer_dz2H · 2023-10-31

**Soundness:** 3 good
**Presentation:** 3 good
**Contribution:** 3 good
**Rating:** 5
**Confidence:** 3

**Summary:**

This paper introduces a stochastic dual gradient descent method for optimizing the Gaussian process posterior computation.

**Strengths:**

The authors present a novel "dual" formulation for the Gaussian process regression problem. After studying the condition number of new and old formulations, the authors observe that the "dual" formulation allows for the use of larger learning rates, indicating its potential to converge faster. They then propose the stochastic dual gradient descent method, leveraging various optimization techniques based on the "dual" formulation, including feature and coordinate sampling (or minibatch) [1], Nesterov's acceleration [2], and Polyak averaging. Notably, the authors introduce a new averaging scheme called geometric averaging.

The paper is overall well-structured, clear, logically presented, and readable. It contains minimal typos and lacks theoretical flaws. Moreover, the authors conduct sufficient numerical experiments to validate the effectiveness of their proposed optimizer.

[1] "Sampling from Gaussian Process Posteriors using Stochastic Gradient Descent"
[2] Y. Nesterov, "A method for unconstrained convex minimization problems with a convergence rate of O(1/k^2)"
[3] B. T. Polyak, "New stochastic approximation type procedures," Avtomatika i Telemekhanika, 1990.

**Weaknesses:**

The authors do not provide a theoretical justification to verify the convergence of the proposed method. Nevertheless, it is likely that convergence can be ensured under mild conditions, as the optimization techniques employed are standard and well-established in the community and literature.

From my perspective, the primary contribution of this paper lies in the introduction of the "dual" formulation, as presented on page 4 after Equation (2). This formulation allows for the use of larger step sizes, which suggests the potential for faster convergence. While the remaining studies and techniques are also important, they are somewhat incremental and standard. Consequently, I am uncertain about whether the paper's contribution alone justifies its publication in ICLR. As a result, I have assigned a boundary score and defer to the area chair's judgment for the final decision on acceptance.

**Questions:**

See weakness.

---

> ### Author Response · Authors · 2023-11-16
> **Thank you for your feedback, please consider our response below!**
>
> Thank you for taking the time to read our work and providing feedback. We are delighted to hear that our paper was structured well and presented clearly and logically! Below we address the key concerns:
>
> -----
> **1. Geometric averaging**
>
> > Notably, the authors introduce a new averaging scheme called geometric averaging.
>
> We want to clarify that we did not introduce geometric averaging, which is widely used in deep learning. **Our contribution is the observation that, for the estimators considered, geometric averaging outperforms arithmetic averaging due to the presence of multiplicative noise**.
> This choice is unorthodox because almost all previous analysis of SGD uses gradient estimators with additive noise. In that setting, arithmetic averaging (as opposed to geometric) is optimal [(Dieuleveut et. al. 2019)](https://arxiv.org/abs/1602.05419).
>
> **2. Theoretical demonstration of convergence**
>
> > The authors do not provide a theoretical justification to verify the convergence of the proposed method. Nevertheless, it is likely that convergence can be ensured under mild conditions, as the optimization techniques employed are standard and well-established in the community and literature.
>
> Our paper is targeted at GP researchers and practitioners. While using SGD for GP inference has recently been found to be very promising ([Lin et al. 2023](https://arxiv.org/pdf/2306.11589.pdf) and [Dai et. al. 2014](https://arxiv.org/pdf/1407.5599.pdf)), **previous work makes a number of suboptimal choices when it comes to the details of the algorithm's formulation. Our contribution is handling these details carefully, highlighting to researchers and practitioners how to make SGD inference even more effective**, particularly on harder problems like the graph kernel comparison.
>
>
> In the **full-batch (non stochastic) case, the analysis of the algorithm is standard** ([Nesterov 1983](https://www.mathnet.ru/php/archive.phtml?wshow=paper&jrnid=dan&paperid=46009&option_lang=eng)), and it is known that the dual objective will result in faster convergence than its primal counterpart due to conditioning. See, for instance, Section 3.7, page 289, of [Bubeck 2015](https://arxiv.org/pdf/1405.4980.pdf), a textbook on convex optimisation.
>
> Our full algorithm makes use of an estimator with multiplicative noise. **The convergence of Stochastic GD under multiplicative noise is currently at the forefront of theoretical research and beyond the scope of our more practical work.** We have extended our discussion of pertinent work in this space in Section 3.4.
>
>
> **3. Significance**
>
> >While the remaining studies and techniques are also important, they are somewhat incremental and standard. Consequently, I am uncertain about whether the paper's contribution alone justifies its publication in ICLR.
>
> We agree that our paper uses insights from the theory literature. Indeed, this is highlighted throughout our text. To our knowledge, these insights have not been studied in a unified way by the kernel/GP community before. Even worse, **all applied kernel/GP work we are familiar with, which use SGD methods, make suboptimal algorithmic design choices**. On this basis we believe that a simple algorithm that combines recent theoretical advancements in a principled way, and  is **the first alternative which reliably outperforms conjugate gradient-based algorithms used in state-of-the-art Gaussian process software packages, can be of value to the ICLR community.**
>
> We would also like to note that in machine learning, there is a long history of combinations of small improvements eventually leading to breakthrough performance: the most prominent example, arguably, is the development of deep neural networks, which required small tweaks to initialisation, activation functions, architecture, optimisation, and other seemingly incremental factors in order to result in today’s effectiveness.

---

> > ### Comment · Reviewer_dz2H · 2023-11-19
> >
> > I appreciate the authors' detailed response to my concerns, and I also take note of the feedback provided by the other reviewers and the author's responses. Considering the concerns also raised by other reviewers about the theoretical justification and the perceived limited novelty of this paper due to the combination of previously well-known techniques, I've decided not to revise my score at this moment.
> >
> > Additionally, Reviewer WMvZ suggested that the authors might consider including comparisons between the proposed stochastic dual gradient descent and other well-known optimizers like AdaGrad or Adam. While Figure 7 presents a comparison among previous methods, I did not observe a direct comparison with the proposed method. The results for the proposed method with varying setups are depicted in Figure 3; however, a direct comparison of the proposed method's efficiency with the others is crucial, especially considering the authors' assertion of faster convergence as the main contribution.

---

> ### Author Response · Authors · 2023-11-19
>
> Thank you for your comment! We would like to clarify **that "Nesterov" in Figure 7 refers to our proposed method**. To prevent further confusion, we will change the corresponding label to "SDD". Further, we would like to reiterate that Figure 7 illustrates that our method reaches solutions with **several magnitudes smaller squared $K$- and $K^2$-norms** compared to AdaGrad, RMSprop and Adam.

---

### Official Review · Reviewer_WMvZ · 2023-11-05

**Soundness:** 2 fair
**Presentation:** 3 good
**Contribution:** 2 fair
**Rating:** 5
**Confidence:** 3

**Summary:**

This paper proposes a stochastic gradient descent method for solving the kernel ridge regression problem. In particular, three aspects are covered: (1) a dual objective that allows a larger learning rate; (2) a stochastic approximation that brings in effective utilization of stochastic gradients; (3) momentum and geometric iterate averaging. By combining these aspects, the algorithm is demonstrated be faster compared to baselines in experiments.

**Strengths:**

* This paper proposes a new method for the kernel ridge regression problem.
* Experimental results show that the proposed algorithms can achieve better performance than baselines. When combined with the Gaussian process, the method can also achieve comparable performance to that of graph neural networks.

**Weaknesses:**

* This paper only provides numerical experiments to evaluate the performance of different algorithms. However, it would be good if rigorous theoretical guarantees could be proved, at least for some special cases. Besides, I think the authors stress too much on the algorithm details, which can be deferred to the appendix for a major part of them while trying to leave some room for theoretical analysis.
* There are many different optimizers for the kernel ridge regression, such as AdaGrad, Adam, etc. The authors should also try these methods in the experiments.
* The algorithm design is a bit incremental to me, as it looks like a combination of standard existing approaches, which is tuned for the specific tasks. Then, the idea of the algorithm design may be difficult to extend to other tasks.
* Besides, it is not clear to me whether the variance of stochastic gradient is really a big issue from Figure 2, as the authors do not add the full-gradient version for comparison. If controlling the variance is important, the authors may also need to consider variance-reduce techniques (e.g., SVRG) and add them to the experiments.

**Questions:**

See the weakness part.

---

> ### Author Response · Authors · 2023-11-16
> **Thank you for your feedback! Please consider our response to your questions (1/2)**
>
> Thank you for your review! We are delighted to see the significance of our graph neural network demonstration pointed out in your review! We would also like to highlight that our method reliably outperforms conjugate gradients, which is the standard optimisation method for this problem. We address your key concerns below.
>
> ----
>
> **1. Theoretical guarantees**
>
> > "This paper only provides numerical experiments... would be good if rigorous theoretical guarantees could be proved, at least for some special cases."
>
> Our paper is targeted at GP researchers and practitioners. While using SGD for GP inference has recently been found to be very promising ([Lin et al. 2023](https://arxiv.org/pdf/2306.11589.pdf) and [Dai et. al. 2014](https://arxiv.org/pdf/1407.5599.pdf)), **previous work makes a number of suboptimal choices** when it comes to the details of the algorithm's formulation. **Our contribution is handling these details carefully, highlighting to researchers and practitioners how to make SGD inference even more effective**, particularly on harder problems like the graph kernel comparison.
>
> In the **full-batch (non-stochastic) case, the analysis of the algorithm is standard** ([Nesterov 1983](https://www.mathnet.ru/php/archive.phtml?wshow=paper&jrnid=dan&paperid=46009&option_lang=eng)), and it is known that the dual objective will result in faster convergence than its primal counterpart due to conditioning. See, for instance section 3.7, page 289, of [Bubeck 2015](https://arxiv.org/pdf/1405.4980.pdf), a textbook on convex optimization.
>
> Our full algorithm makes use of an estimator with multiplicative noise. **The convergence of Stochastic GD under multiplicative noise is currently at the forefront of theoretical research and beyond the scope of our more practical work.** We have extended our discussion of pertinent work in this space in Section 3.4. We hope that our work's demonstration of the empirical performance achievable with SGD under multiplicative noise will motivate research in this area by the optimisation community.
>
> **2. Algorithmic details.**
>
> > "The authors stress too much on the algorithm details. which can be deferred to the appendix for a major part of them while trying to leave some room for theoretical analysis."
>
> Thank you for this comment. **Our aim was to show how to make the already promising SGD-based learning algorithms for Gaussian processes more effective by giving a careful treatment to algorithmic details. To do so, it is very important to precisely state the algorithm we recommend**, particularly for readers coming from the Gaussian process rather than optimisation community.
>
> At the same time, we recognize that papers which focus more on the side of theoretical analysis are also valuable. We believe it is best to have both kinds of papers available in the literature, so that readers can more precisely find what they are looking for.
>
> **3. Comparison with other optimisers.**
>
> >  different optimizers such as AdaGrad, Adam, etc. The authors should also try these methods.
>
> We now include an empirical comparison on the POL dataset of AdaGrad vs RMSprop vs Adam vs our method. **We show that our method compares favourably to these methods by a significant margin: see [linked figure](https://anonymous.4open.science/r/SDD-GPs-5486/rebuttal_plots/optimiser_comparison_pol.pdf) and Appendix C.2.**
>
> Thank you for pointing this comparison out. We originally opted to omit these algorithms, as they are usually designed to tackle problems with a non-constant curvature. GP regression has a constant curvature: it yields a quadratic objective where Nesterov-type momentum is theoretically rate-optimal ([Dieuleveut et. al. 2017](https://arxiv.org/abs/1602.05419) and [Jain et. al. 2018](https://arxiv.org/abs/1704.08227)). However, given the prevalence of other optimisation algorithms, we agree that a comparison to them is valuable to the larger community!
>
> (continued in next comment)

---

> > ### Author Response · Authors · 2023-11-16
> > **Thank you for your feedback! Please consider our response to your questions (2/2)**
> >
> > **4. Significance of work**
> >
> > > The algorithm design is a bit incremental... tuned for the specific tasks. ...may be difficult to extend to other tasks.
> >
> > We would like to highlight that **Gaussian processes are a very important class of models, which are widely used for Bayesian optimisation in areas like computational chemistry** (for instance in protein design, see [here](https://pubs.acs.org/doi/10.1021/acscentsci.7b00572)), **hyperparameter tuning** (see for instance [Google Vizier](https://github.com/google/vizier)), and others. Designing specialised optimisation algorithms which take advantage of their specific structure is therefore a worthwhile problem, which has long been held back by cubic computational cost in data set size. Building on the initial results of [Lin et al. 2023](https://arxiv.org/pdf/2306.11589.pdf), **our algorithm is the first alternative which reliably outperforms conjugate gradient-based algorithms used in state-of-the-art Gaussian process software packages.**
> >
> > Extending this, solving quadratic problems emerging from kernel regression efficiently can also be applied to inference in non-conjugate settings, like classification. For instance, [Antoran et. al. 2023](https://arxiv.org/pdf/2210.04994.pdf) combine SGD with the Laplace approximation to perform inference with the NTK on ImageNet classification. Our faster dual SGD technique is a drop-in replacement into their paper.
> >
> > **5. Combination of existing approaches**
> >
> > >The algorithm design is ... a combination of standard existing approaches
> >
> > We agree that our paper uses insights from the theory literature—indeed, this is highlighted throughout our text. To our knowledge, these insights have not been studied in a unified way by the kernel/GP community before. Even worse, **all applied kernel/GP work we are familiar with, which use SGD methods, make suboptimal algorithmic design choices**. On this basis we believe that a simple algorithm that combines recent theoretical advancements in a principled way and produces state-of-the-art results is valuable to the ICLR community.
> >
> > We would also like to note that in machine learning, there is a long history of combinations of small improvements eventually leading to breakthrough performance: the most prominent example, arguably, is the development of deep neural networks, which required small tweaks to initialisation, activation functions, architecture, optimisation, and other seemingly incremental factors in order to result in today’s effectiveness.
> >
> > **6. Gradient variance**
> >
> > > ...it is not clear to me whether the variance of stochastic gradient is really a big issue from Figure 2, as the authors do not add the full-gradient version for comparison.
> >
> > Gradient variance can significantly harm SGD performance if proper precautions are not taken. This can be seen through the effect of geometric averaging, which aims to ameliorate the effects of noisy gradient estimates. This is illustrated **in Figure 3, where the blue line with averaging (i.e. variance reduction) is consistently below the non-averaged olive-green line.**
> >
> > This can also be seen when using an estimator of the gradient with additive (instead of multiplicative) noise. With geometric averaging and our proposed multiplicative noise gradient estimator, the effects of variance are notably lessened. **We have added ablations over step-size and batch size for both additive and multiplicative estimators of the gradient (see figures [here](https://anonymous.4open.science/r/SDD-GPs-5486/rebuttal_plots/lr_comparison_pol.pdf) and [here](https://anonymous.4open.science/r/SDD-GPs-5486/rebuttal_plots/batch_size_comparison_pol.pdf) and Appendix C.3 and C.4 for further details).** The additive estimator's performance suffers when the step-size is increased, or batch size is decreased; that is when the gradient estimator's variance increases. In contrast, the proposed multiplicative estimator's performance increases as the step-size increases, whereas the additive estimator gets worse. Similarly, the multiplicative estimator is significantly more robust to reductions in batch size.
> >
> > ---
> >
> > **Summarising**: we hope we have addressed all your points through our additional experiments and clarifications! We thank you for the actionable feedback. These have helped us realise points that need to be communicated better in our paper, and we have revised the manuscript in light of that.

---

### Author Response · Authors · 2023-11-20
**Summary**

We would like to again thank the reviewers for their time reading our work and for all of their suggestions. Here, we summarise our **new experiments** and our response to WMvZ, dz2H and 1x4h's shared **concerns regarding significance**.

----

* **Comparison with Adam, RMSProp, AdaGrad** is available [here (our method is 'Nesterov' in legend)](https://anonymous.4open.science/r/SDD-GPs-5486/rebuttal_plots/optimiser_comparison_pol.pdf) and in Appendix C.2. These methods are designed for non-quadratic problems and underperform regular SGD+nestervov when the loss curvature is constant (our setting).



* **Learning rate and batch size ablations** for both additive and multiplicative gradient estimators are available [here](https://anonymous.4open.science/r/SDD-GPs-5486/rebuttal_plots/lr_comparison_pol.pdf) and [here](https://anonymous.4open.science/r/SDD-GPs-5486/rebuttal_plots/batch_size_comparison_pol.pdf), and in Appendix C.3.
    *  We have also included [here](https://anonymous.4open.science/r/SDD-GPs-5486/rebuttal_plots/primal_lr_divergence_pol_10.pdf) and [here](https://anonymous.4open.science/r/SDD-GPs-5486/rebuttal_plots/primal_lr_divergence_pol_20.pdf), which appear as Figures 12 and 13 in Appendix C.6, a demonstration of how primal SGD (i.e. an additive noise estimator) diverges at moderate learning rates.


* **Kernel hyperparameter learning**: We have emphasized that our paper tackles "the fixed kernel setting" in our introduction.
    * We have added a demonstration of how the posterior samples obtained with our method can be used to learn hyperparameters (using the standard Hutchinson trace estimator of the logdet term) in Appendix C.7 and [here](https://anonymous.4open.science/r/SDD-GPs-5486/rebuttal_plots/mll_optim.pdf).


* **Ease of implementation**: This [Jupyter Notebook](https://anonymous.4open.science/r/SDD-GPs-5486/sdd.ipynb) samples from a GP posterior with our method in 20 lines of numpy.


* **Beautiful visualisations**: Figure [here](https://anonymous.4open.science/r/SDD-GPs-5486/rebuttal_plots/toy_comparison_rebuttal.pdf) in Appendix C.1 qualitatively compares our SGD posterior samples to those from CG and SGVI.


* **Robustness to ill-conditioning**: [This](https://anonymous.4open.science/r/SDD-GPs-5486/rebuttal_plots/ill_conditioned_pol.pdf) figure has been added to Appendix C.5 showing SGD's robustness to setting the observation noise variance to $10^{-6}$.


* **Instability in blue curve in (old) figure 2**. This was due to instability of float32 when the error is very small ($10^{-7}$). We re-made the plot with float64 [here](https://anonymous.4open.science/r/SDD-GPs-5486/rebuttal_plots/batch_variants_pol.pdf).


---


### Significance of the work

* The methods presented in our paper have appeared before in the optimisation theory literature (and our text is clear about this!)
* They have not been adopted by the Gaussian process or linear modelling community, where suboptimal choices consistently leave performance on the table.
    * e.g. [Dai et. al. 2014](https://arxiv.org/pdf/1407.5599.pdf), [Antoran et. al. 2023](https://arxiv.org/pdf/2210.04994.pdf) and [Lin et al. 2023](https://arxiv.org/pdf/2306.11589.pdf) all use additive noise estimators.
    * The empirical superiority of multiplicative noise estimators is a recent and quite niche development that makes a big difference!
* An algorithm that can be implemented in 20 lines of numpy [(Notebook)](https://anonymous.4open.science/r/SDD-GPs-5486/sdd.ipynb) and produces state-of-the-art results in GP regression and Bayesian optimisation can be of interest to ICLR community.

---

### Meta-Review · Area_Chair_ZQYC · 2023-12-05

**Metareview:**

This paper presents a stochastic gradient descent method for GP, introducing a dual objective allowing for larger learning rates, effective use of stochastic gradients, and a combination of momentum and geometric iterate averaging. The authors address significant concerns raised by reviewers, notably regarding the significance of the work, comparisons with other optimizers, and theoretical guarantees. They provide additional experiments, comparisons, and clarifications, highlighting the empirical superiority of their approach and its practicality for Gaussian process (GP) researchers and practitioners. While some reviewers noted the lack of theoretical guarantees for convergence, the practical relevance and empirical results justify its contribution, and its acceptance.

**Justification For Why Not Higher Score:**

Lack of new theoretical insights

**Justification For Why Not Lower Score:**

Practical and effective approach for GP regression and Bayesian optimization.

---

### Decision · Program_Chairs · 2024-01-16

Accept (poster)